# EXPLANATION UNCERTAINTY
# WITH DECISION BOUNDARY AWARENESS

## ABSTRACT

Post-hoc explanation methods have become increasingly depended upon for understanding black-box classifiers in high-stakes applications, precipitating a need for reliable explanations. While numerous explanation methods have been proposed, recent works have shown that many existing methods can be inconsistent or unstable. In addition, high-performing classifiers are often highly nonlinear and can exhibit complex behavior around the decision boundary, leading to brittle or misleading local explanations. Therefore, there is an impending need to quantify the uncertainty of such explanation methods in order to understand when explanations are trustworthy. We introduce a novel uncertainty quantification method parameterized by a Gaussian Process model, which combines the uncertainty approximation of existing methods with a novel geodesic-based similarity which captures the complexity of the target black-box decision boundary. The proposed framework is highly flexible; it can be used with any black-box classifier and feature attribution method to amortize uncertainty estimates for explanations. We show theoretically that our proposed geodesic-based kernel similarity increases with the complexity of the decision boundary. Empirical results on multiple tabular and image datasets show that our decision boundary-aware uncertainty estimate improves understanding of explanations as compared to existing methods.

## 1 INTRODUCTION

Machine learning models are becoming increasingly prevalent in a wide variety of industries and applications. In many such applications, the best performing model is opaque; post-hoc explainability methods are one of the crucial tools by which we understand and diagnose the model's predictions. Recently, many explainability methods, termed *explainers*, have been introduced in the category of local feature attribution methods. That is, methods that return a real-valued score for each feature of a given data sample, representing the feature's relative importance with respect to the sample prediction. These explanations are local in that each data sample may have a different explanation. Using local feature attribution methods therefore helps users better understand nonlinear and complex black-box models, since these models are not limited to using the same decision rules throughout the data distribution.

Recent works have shown that existing explainers can be inconsistent or unstable. For example, given similar samples, explainers might provide different explanations (Alvarez-Melis & Jaakkola, 2018; Slack et al., 2020). When working in high-stakes applications, it is imperative to provide the user with an understanding of whether an explanation is reliable, potentially problematic, or even misleading. A way to guide users regarding an explainer's reliability is to provide corresponding *uncertainty quantification* estimates.

One can consider explainers as function approximators; as such, standard techniques for quantifying the uncertainty of estimators can be utilized to quantify the uncertainty of explainers. This is the strategy utilized by existing methods for producing uncertainty estimates of explainers (Slack et al., 2021; Schwab & Karlen, 2019). However, we observe that for explainers, this is not sufficient; because in addition to uncertainty due to the function approximation of explainers, explainers also have to deal with the uncertainty due to the complexity of the decision boundary (DB) of the black-box model in the local region being explained.

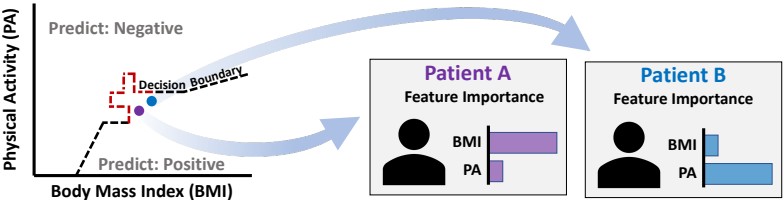

Figure 1: Illustrative example of how similar data samples can result in very different feature importance scores given a black-box model with nonlinear decision boundary. Here, two similar patients with similar predictions are given opposing feature importance scores which could result in misguided recommendations. We define a similarity based on the geometry of the decision boundary between any two given samples (red line). While the two patients are close together in the Euclidean sense, they are dissimilar under the proposed WEG kernel similarity. Using GPEC would return a high uncertainty measure for the explanations, which would flag the results for further investigation.

Consider the following example: we are using a prediction model for a medical diagnosis using two features, level of physical activity and body mass index (BMI) (Fig. 1). In order to understand the prediction and give actionable recommendations to the patient, we use a feature attribution method to evaluate the relative importance of each feature. Because of the nonlinearity of the prediction model, patients A and B show very similar symptoms, but are given very different recommendations. Note that while this issue is related to the notion of explainer uncertainty, measures of uncertainty that only consider the explainer would not capture this phenomenon. This suggests that any notion of uncertainty is incomplete without capturing information related to the local behavior of the model. Therefore, the ability to quantify uncertainty for DB-related explanation instability is desirable.

We approach this problem from the perspective of similarity: given two samples and their respective explanations, how closely related should the explanations be? From the previous intuition, we define this similarity based on a geometric perspective of the DB complexity between these two points. Specifically, we propose a novel geodesic-based kernel similarity metric, which we call the Weighted Exponential Geodesic (WEG) kernel. The WEG kernel encodes our expectation that two samples close in Euclidean space may not actually be similar if the DB within a local neighborhood of the samples is highly complex.

Using our similarity formulation, we propose the **G**aussian **P**rocess **E**xplanation Un**C**ertainty (GPEC) framework, which is an instance-wise, model-agnostic, and explainer-agnostic method to quantify the uncertainty of explanations. The proposed notion of uncertainty is complementary to existing quantification methods. Existing methods primarily estimate the uncertainty related to the choice in model parameters and fitting the explainer, which we call *function approximation uncertainty*, and does not capture uncertainty related to the DB. GPEC can combine the DB-based uncertainty with function approximation uncertainty derived from any local feature attribution method.

In summary, we make the following contributions:

- We introduce a geometric perspective on capturing explanation uncertainty and define a novel geodesic-based similarity between explanations. We prove theoretically that the proposed similarity captures the complexity of the decision boundary from a given black-box classifier.

- We propose a novel Gaussian Process-based framework that combines A) uncertainty from decision boundary complexity and B) explainer-specific uncertainty to generate uncertainty estimates for any given feature attribution method and black box model.

- Empirical results show GPEC uncertainty improves understanding of feature attribution methods.

## 2 RELATED WORKS

**Explanation Methods.** A wide variety of methods have been proposed for the purpose of improving transparency for pre-trained black-box prediction models (Guidotti et al., 2018; Barredo Arrieta et al., 2020). Within this category of *post-hoc* methods, many methods focus on *local* explanations, that is, explaining individual predictions rather than the entire model. Some of these methods generate explanations through local *feature selection* (Chen et al., 2018; Masoomi et al., 2020). In this

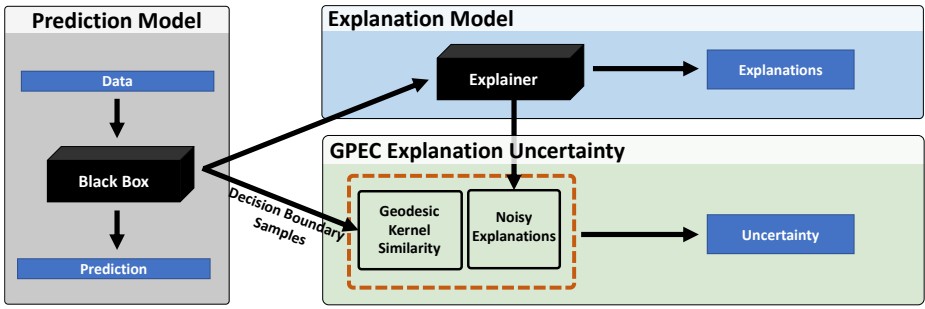

Figure 2: Schematic for the GPEC uncertainty estimation process. GPEC can be used in conjunction with a black-box classifier and explainer to derive an estimate of explanation uncertainty. GPEC takes samples from classifier's decision boundary plus (possibly noisy) explanations from the explainer and fits a Gaussian Progress Regression model with Weighted Exponential Geodesic Kernel. The variance of the predictive distribution combines uncertainty from the black-box classifier complexity and the approximation uncertainty from explainer.

work, we focus primarily on *feature attribution* methods, which return a real-valued score for each feature in the sample. For example, LIME (Ribeiro et al., 2016) trains a local linear regression model to approximate the black-box model. Lundberg & Lee (2017) generalizes LIME and five other feature attribution methods using the SHAP framework, which fulfill a number of desirable axioms. While some methods such as LIME and its SHAP variant, KernelSHAP, are model-agnostic, others are designed for specific model architectures, such as neural networks (Bach et al., 2015; Shrikumar et al., 2017; Sundararajan et al., 2017; Erion et al., 2021), tree ensembles (Lundberg et al., 2020), or Bayesian neural networks (Bykov et al., 2020), taking advantage of those specific architectures. Another class of method involves the training of separate surrogate models to explain the black-box model (Dabkowski & Gal, 2017; Chen et al., 2018; Schwab & Karlen, 2019; Guo et al., 2018; Jethani et al., 2022). Once trained, surrogate-based methods are typically very fast; explanation generation generally only requires a single inference step from the surrogate model.

**Explanation Uncertainty.** One option for improving the trustworthiness of explainers is to quantify the associated explanation uncertainty. Bootstrap resampling techniques have been proposed as a way to estimate uncertainty from surrogate-based explainers (Schwab & Karlen, 2019; Schulz et al., 2022). Guo et al. (2018) also proposes a surrogate explainer parameterized with a Bayesian mixture model. Alternatively, Bykov et al. (2020) and Patro et al. (2019) introduce methods for explaining Bayesian neural networks, which can be transferred to non-Bayesian neural networks. Covert & Lee (2021) derives an unbiased version of KernelSHAP and investigates an efficient way of estimating its uncertainty. Zhang et al. (2019) categorizes different sources of variance in LIME estimates. Several methods also investigate LIME and KernelSHAP in a Bayesian context; for example calculating a posterior over attributions (Slack et al., 2021), investigating the use of priors for explanations (Zhao et al., 2021), or using active learning to improve sampling (Saini & Prasad, 2022).

However, all existing methods for quantifying explanation uncertainty only consider the uncertainty of the explainer as a function approximator. This work introduces an additional notion of uncertainty for explainers that takes into account the uncertainty of the explainer due to the DB.

## 3 UNCERTAINTY QUANTIFICATION FOR BLACK-BOX EXPLAINERS

We now outline the GPEC framework, which we parameterize with a Gaussian Process (GP) regression model. We define a vector-valued GP which is trained on data samples as input and explanations as labels. More concretely, consider a data sample $x \in \mathcal{X} \subset \mathbb{R}^d$ that we want to explain in the context of a black-box prediction model $F : \mathcal{X} \to [0, 1]$, in which the output corresponds to the probability for the positive class. For simplicity, we consider the binary classifier case; we extend to the multiclass case in Section C.3. We apply any given feature attribution method $E : \mathcal{X} \to \mathbb{R}^s$, where $s$ is the dimension of the explanations $e = E(x)$.

Let us draw samples $x_1, \ldots, x_M$ from the data distribution and generate their respective explanations $e_1, \ldots, e_M$. We assume that each explanation $e_m$ is generated from an unobserved latent function $\mathcal{E}$

plus some independent Gaussian noise $\eta_m$.

$$e_m = \mathcal{E}(x_m) + \eta_m \quad s.t. \quad \underbrace{\mathcal{E}(x_m) \sim \mathcal{GP}(0, k(x, x'))}_{\text{Decision Boundary-Aware Uncertainty}} \quad \underbrace{\eta_m \sim \mathcal{N}(0, \tau_m^{-1})}_{\text{Function Approximation Uncertainty}} \tag{1}$$

where $k(\cdot, \cdot)$ is the specified kernel function and $\tau_m > 0$. We decompose each explanation into two components, $\mathcal{E}(x_m)$ and $\eta_m$, which represent two separate sources of uncertainty: 1) a *decision boundary-aware* uncertainty which we capture using the kernel similarity, and 2) a *function approximation* uncertainty from the explainer. After specifying $\mathcal{E}(x_m)$ and $\eta_m$, we can combine the two sources by calculating the posterior predictive distribution for the GP model; we take the 95% confidence interval of this distribution to be the GPEC uncertainty estimate. Due to space constraints, we provide an overview of GP regression in App. B.2.

**Function Approximation Uncertainty.** The $\eta_m$ component of Eq. 1 represents the uncertainty stemming from explainer specification. For example, this uncertainty can be captured from increased variance from undersampling (perturbation-based explainers) or from the increased bootstrap resampling variance from model misspecification (surrogate-based explainers).

We can therefore define $\tau$ based on the chosen explainer. Explainers that include some estimate of uncertainty (e.g. BayesLIME, BayesSHAP, CXPlain) can be directly used to estimate $\tau$. For example, CXPlain returns a distribution of feature importance values for a sample $x_m$; the variance of this distribution can be directly used as the variance of $\eta_m$. For other stochastic explanation methods that do not explicitly estimate uncertainty, we can estimate $\tau$ empirically by resampling explanations for the same data sample:

$$\hat{\tau}_m = [\frac{1}{|K|} \sum_{i=1}^{K} (E_i(x_m) - \bar{E}(x_m))^2]^{-1} \quad s.t. \quad \bar{E}(x_m) = \frac{1}{|K|} \sum_{i=1}^{K} E_i(x_m) \tag{2}$$

where each $E_i(x_m)$ is a sampled explanation for $x_m$. Alternatively, for deterministic explanation methods we can omit the noise and assume that the generated explanations are noiseless.

**Decision Boundary-Aware Uncertainty.** In contrast, the $\mathcal{E}(x_m)$ component of Eq. 1 draws possible functions from the GP prior that could have generated the observed explanations, with the function space defined by the choice of kernel. Intuitively, the kernel encodes our *a priori* definition of how similar two explanations should be given the similarity of the inputs. In other words, how much information do we expect a given point $x$ to provide for a nearby point $x'$ with respect to their explanations? We use this kernel specification to define a source of uncertainty dependent on the behavior of nearby explanations. In particular, we consider a novel kernel formulation that reflects the complexity of the DB in a local neighborhood of the samples; this is detailed in Section 4. For any given kernel, we can interpret the distribution of possible functions $\mathcal{E}(x_m)$ as an estimate of uncertainty based on the set of previously observed or sampled explanations.

## 4 WEIGHTED EXPONENTIAL GEODESIC KERNEL

Intuitively, the GP kernel encodes the assumption that each individual explanation gives some information about the explanations around it; the choice of kernel defines the neighborhood and magnitude of this shared information. In the GP framework introduced in Section 3, the kernels define the relationship or similarity between two explanations $\mathcal{E}(x)$ and $\mathcal{E}(x')$ based soley on some function of their inputs $x$ and $x'$. For example, stationary kernels assign the same similarity for any two inputs regardless of where in the data manifold they are located. Instead, we want to encode the assumption that when the DB is complex, knowing an explanation $\mathcal{E}(x)$ gives limited information about other nearby explanations (see Fig. 3).

### 4.1 GEOMETRY AND GEODESICS OF THE DECISION BOUNDARY

We want to relate kernel similarity to the behavior of the black-box model; specifically, the complexity or smoothness of the DB. Given any two points on the DB, the relative complexity of the boundary segment between them can be approximated by the segment length. The simplest form that the DB can take is a linear boundary connecting the two points; this is exactly the minimum distance between the points. As the complexity of the DB grows, there is a general corresponding increase in segment length.

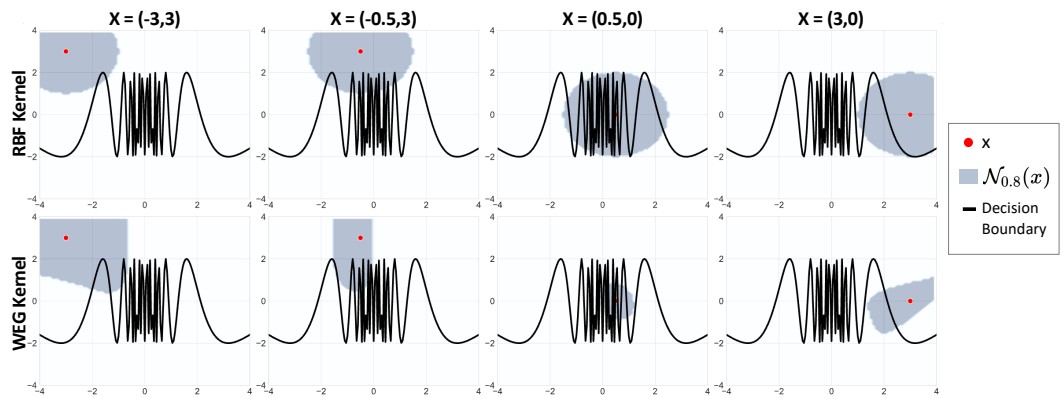

Figure 3: Comparison of RBF and WEG kernel similarity neighborhoods. The gray highlighted region $\mathcal{N}_{0.8}(x) = \{x' : k(x, x') \geq 0.8\}$ indicate points where the kernel similarity is greater than 0.8 with respect to the red point $x$. The black line is the decision boundary for the classifier $f(x_1, x_2) = 2\cos(\frac{10}{x_1}) - x_2$, where $f(x_1, x_2) \geq 0$ indicates class 1, and $f(x_1, x_2) < 0$ indicates class 0. We see that for GPEC, $\mathcal{N}_{0.8}(x)$ shrinks as $x$ moves closer to segments of the decision boundary which are more complex.

Concretely, we define the DB as a $(d-1)$-dimensional Riemannian manifold embedded in $\mathbb{R}^d$. Given the binary classifier $F$, we define $\mathcal{M} = \{x \in \mathbb{R}^d : F(x) = \frac{1}{2}\}$ representing the DB[1] of $F$. Given this interpretation, we can define distances along the DB as geodesic distances in $\mathcal{M}$.

$$d_{geo}(m, m') = \min_{\gamma} \int_0^1 ||\gamma(t)|| \mathrm{d}t \quad \forall m, m' \in \mathcal{M} \tag{3}$$

The mapping $\gamma : [0, 1] \rightarrow \mathcal{M}$, defined such that $\gamma(0) = m$ and $\gamma(1) = m'$, is a parametric representation of a 1-dimensional curve on $\mathcal{M}$. We can adapt geodesic distance in our kernel selection through the exponential geodesic (EG) kernel (Feragen et al., 2015), which is a generalization of the Radial Basis Function (RBF) kernel substituting $\ell_2$ distance with geodesic distance:

$$k_{EG}(x, x') = \exp(-\lambda d_{geo}(x, x')) \qquad k_{RBF}(x, x') = \exp(-\lambda ||x - x'||_2^2) \tag{4}$$

The EG kernel has been previously investigated in the context of Riemannian manifolds (Feragen et al., 2015; Feragen & Hauberg, 2016). In particular, while prior work shows that the EG kernel fails to be positive definite for all values of $\lambda$ in non-Euclidean space, there exists large intervals of $\lambda > 0$ for the EG kernel to be positive definite. Appropriate values can be selected through grid search and cross validation; we assume that a valid value of $\lambda$ has been selected.

Note that the manifold we are interested in is the DB; applying the EG kernel on the data manifold (i.e. directly using $k_{EG}$ in the GPEC formulation) would not capture model complexity. In addition, naïvely using the EG kernel on the DB manifold would not capture the local complexity with respect to a given explanation; the similarity would be invariant to observed explanations. We therefore need to relate the geodesic distances on $\mathcal{M}$ to samples in the data space $\mathcal{X}$.

## 4.2 WEIGHTING THE DECISION BOUNDARY

Consider a probability distribution $p(M)$ with its support defined over $\mathcal{M}$. We weight $p(M)$ according to the $\ell_2$ distance between $M$ and a fixed data sample $x$:

$$q(M|x, \rho) \propto \exp[-\rho ||x - M||_2^2] p(M) \tag{5}$$

We evaluate the kernel function $k(x, x')$ by taking the expected value over the weighted distribution.

$$k_{WEG}(x, x') = \int \int \exp[-\lambda d_{geo}(m, m')] \, q(m|x, \rho) \, q(m'|x', \rho) \, \mathrm{d}m\mathrm{d}m' \tag{6}$$

---

[1]Without loss of generality, we assume that the decision rule for the classifier is set to be $\frac{1}{2}$

Our formulation is an example of a marginalized kernel (Tsuda et al., 2002): a kernel defined on two observed samples $x, x'$ and taking the expected value over some hidden variable. Given that the underlying EG kernel is positive definite, it follows that the WEG kernel forms a valid kernel.

To evaluate the WEG kernel, we theoretically investigate two properties of the kernel. Theorem 1 shows that the WEG kernel is an extension of the EG kernel for data samples not directly on the DB.

**Theorem 1.** *Given two points $x, x' \in \mathcal{M}$, then $\lim_{\lambda \to \infty} k_{WEG}(x, x') = k_{EG}(x, x')$*

Proof details are in App. C.1. Intuitively, as we increase $\lambda$, the manifold distribution closest to the points $x, x'$ becomes weighted increasingly heavily. At the limit, the weighting becomes concentrated entirely on the points $x, x'$ themselves, which recovers the EG kernel. Therefore we see that the WEG kernel is simply a weighting of the EG kernel, which is controlled by $\lambda$.

Theorem 2 establishes the inverse relationship between DB complexity and WEG kernel similarity. Given a black-box model with a piecewise linear DB, we show that this DB represents a local maximum with respect to WEG kernel similarity; i.e. as we perturb the DB to be nonlinear, kernel similarity decreases. We first define *perturbations* on DB. Note that int$(S)$ indicates the interior of a set $S$, f$|_\mathcal{S}$ represents the function $f$ restricted to $S$, and id indicates the identity mapping.

**Definition 1.** Let $\{U_\alpha\}_{\alpha \in I}$ be charts of an atlas for a manifold $\mathcal{P} \subset \mathbb{R}^d$, where $I$ is a set of indices. Let $\mathcal{P}$ and $\widetilde{\mathcal{P}}$ be differentiable manifolds embedded in $\mathbb{R}^d$, where $\mathcal{P}$ is a Piecewise Linear (PL) manifold. Let $R : \mathcal{P} \to \widetilde{\mathcal{P}}$ be a diffeomorphism. We say $\widetilde{\mathcal{P}}$ is a *perturbation* of $\mathcal{P}$ on the $i^{\text{th}}$ chart if $R$ satisfies the following two conditions. 1) There exists a compact subset $K_i \subset U_i$ s.t. $R|_{\mathcal{P} \setminus \text{int}(K_i)} = \text{id}|_{\mathcal{P} \setminus \text{int}(K_i)}$ and $R|_{\text{int}(K_i)} \neq \text{id}|_{\text{int}(K_i)}$. 2) There exists a linear homeomorphism between an open subset $\widetilde{U_i} \subseteq U_i$ with $\mathbb{R}^{d-1}$ which contains $K_i$.

**Theorem 2.** *Let $\mathcal{P}$ be a $(d{-}1)$-dimension PL manifold embedded in $\mathbb{R}^d$. Let $\widetilde{\mathcal{P}}$ be a perturbation of $\mathcal{P}$ and define $\tilde{k}(x, x')$ and $k(x, x')$ as the WEG kernel defined on $\widetilde{\mathcal{P}}$ and $\mathcal{P}$ respectively. Then $\tilde{k}(x, x') < k(x, x') \; \forall x, x' \in \mathbb{R}^d$.*

Proof details are in App. C.2. Theorem 2 implies that, for any two fixed points $x, x'$, their kernel similarity $k_{WEG}(x, x')$ decreases as the black-box DB complexity increases. Within GPEC, the explanations for $x, x'$ become less informative for other nearby explanations and induce a higher explanation uncertainty estimate.

To improve the interpretation of the WEG kernel, we apply a normalization to scale the similarity values to be between $[0, 1]$. We construct the normalized kernel $k^*_{WEG}$ as follows:

$$k^*_{WEG}(x, x') = \frac{k_{WEG}(x, x')}{\sqrt{k_{WEG}(x, x) k_{WEG}(x', x')}} \tag{7}$$

### 4.3 WEG KERNEL APPROXIMATION

In practice, the integral in Eq. 6 is intractable; we approximate the expected value using Monte Carlo (MC) sampling with $K$ samples:

$$k_{WEG}(x, x') \approx \frac{1}{Z_m Z_{m'} K^2} \sum_{i=1}^{K} \sum_{j=1}^{K} \exp[-\lambda d_{geo}(m_i, m_j)] \exp[-\rho(||x - m_i||_2^2 + ||x' - m_j||_2^2)] \tag{8}$$

$Z_m$, $Z_{m'}$ are the normalization constants for $q(M|x, \rho)$ and $q(M'|y, \rho)$, respectively. We can similarly estimate these values using MC sampling:

$$Z_m = \int \exp[-\rho||x - m||_2^2] p(m) \, \mathrm{d}m \approx \frac{1}{K} \sum_{i=1}^{K} \exp[-\rho||x - m_i||_2^2] \tag{9}$$

**GPEC Implementation Details.** Due to space constraints, the full GPEC training algorithm is detailed in Alg. 1 in App. D. To use the GPEC framework, we first train the GP regression model with a set of training samples. Once the GP is trained, we can derive uncertainty estimates for test samples from the 95% confidence interval of the posterior predictive distribution. The WEG kernel requires additional setup time to use in the GP. We first calculate the EG kernel matrix by sampling the black-box DB, then weigh each element of the matrix per Eq. 5. The estimation of

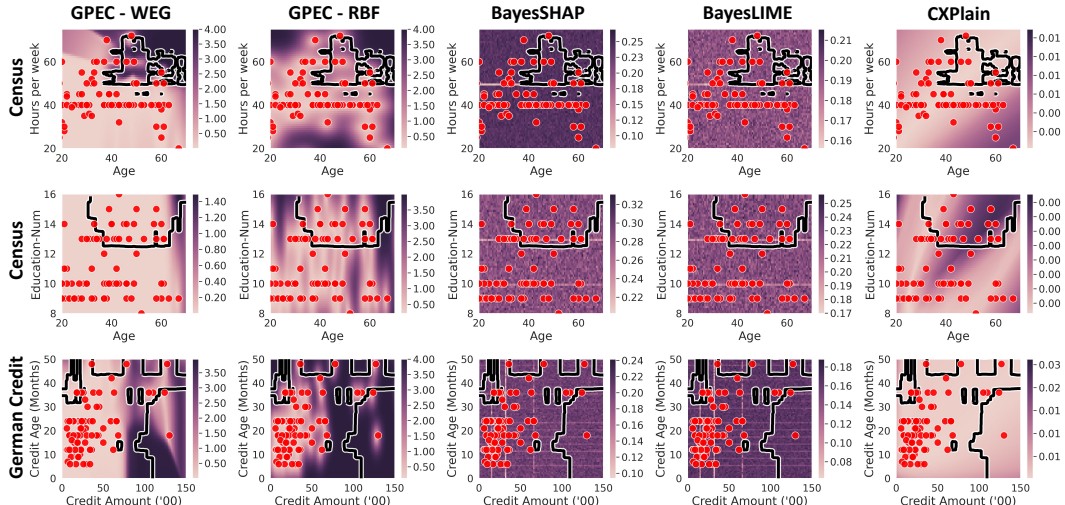

Figure 4: Visualization of estimated uncertainty of explanations for different models and competing methods. The heatmap represents level of uncertainty for a grid of explanations for the feature on the x-axis; darker areas represent higher uncertainty. Red points represent training samples for GPEC and CXPlain as well as the reference samples for BayesSHAP and BayesLIME. The black line represents the black-box DB. The heatmap shows that GPEC-WEG is the only method that captures uncertainty from the DB, due to the WEG kernel similarity formulation.

a classifier's DB is an ongoing area of research; a number of methods have been proposed in the literature (e.g. Yan & Xu (2008); Karimi et al. (2019)). Once GPEC is trained, training cost is amortized during inference; estimating uncertainty for test samples using a GP generally has time complexity of $\mathcal{O}(n^3)$, which can be reduced to $\mathcal{O}(n^2)$ using BBMM (Gardner et al., 2018), and further with variational methods (e.g., Hensman et al. (2015)).

## 5 EXPERIMENTS

We evaluate how well GPEC captures 1) DB-aware uncertainty and 2) functional approximation uncertainty on a variety of datasets and classifiers. In section 5.2 we compare the DB-aware uncertainty and approximation uncertainty components of GPEC. Section 5.3 evaluates how GPEC captures DB complexity. Section 5.4 evaluates how well GPEC combines sources of uncertainty. Due to space constraints we have additional results in the appendix, including sensitivity analysis for GPEC parameters (F.5), execution time comparison F.1, and additional experiments on combining uncertainty (F.3). All experiments were run on an internal cluster using AMD EPYC 7302 16-Core processors, and all source code will be made public.

### 5.1 EXPERIMENTAL SETUP

**Datasets and Models.** Experiments are performed on three tabular datasets (Census, Online Shoppers (Sakar et al., 2019), German Credit) from the UCI data repository (Dua & Graff, 2017) and two image datasets (MNIST (LeCun & Cortes, 2010) and fashion-MNIST (f-MNIST) (Xiao et al., 2017)). GPEC can be used with any black-box model; for our experiments we use XGBoost (Chen & Guestrin, 2016) with log-loss for the tabular datasets and a 4-layer Multi-Layered Perception (MLP) model for the image datasets. Additional dataset details are outlined in App. E.1.

**Implementation Details.** For comparison purposes, we consider GPEC using two different kernels: GPEC-WEG (WEG kernel) and GPEC-RBF (RBF Kernel). Unless otherwise stated, we choose $\lambda = 1.0$ and $\rho = 0.1$ (see App. F.5 for experiments on parameter sensitivity). For the Uncertainty Visualization (Sec. 5.2) and Regularization Test (Sec. 5.3) we use GPEC with the KernelSHAP explainer. We train the GP parameterizing GPEC with BBMM (Gardner et al., 2018). Samples from the DB are drawn using DBPS (Yan & Xu, 2008) for tabular datasets and DeepDIG (Karimi et al., 2019) for image datasets. Geodesic distances are estimated using ISOMAP Tenenbaum et al. (2000). Additional implementation details are available in App. D.

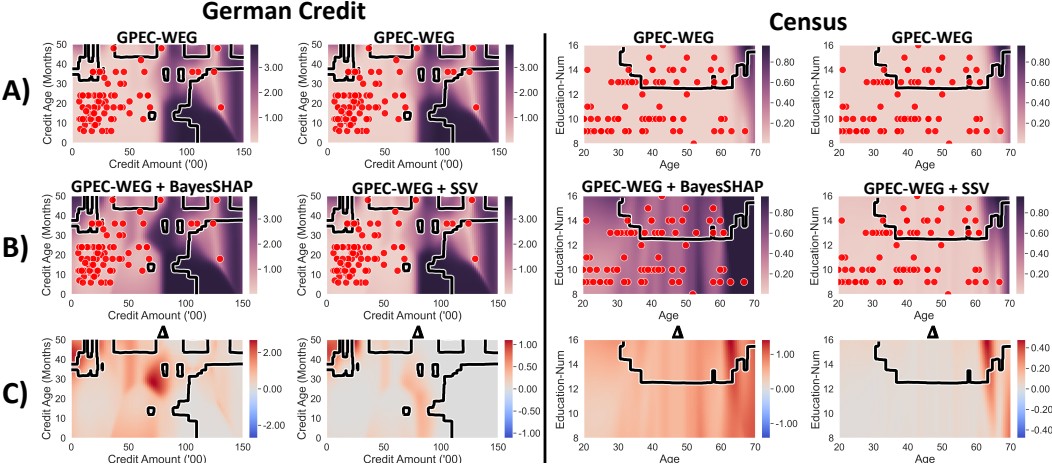

Figure 5: Evaluation of GPEC's ability to combine DB-aware uncertainty and functional approximation uncertainty. Row (A) visualizes the GPEC-WEG uncertainty estimate using only DB-aware uncertainty. Row (B) combines the DB-aware uncertainty in (A) with the functional approximation uncertainty from the two explainers: BayesSHAP and SSV. Row (C) visualizes the change in uncertainty estimate between (A) and (B).

**Competing Methods.** We compare GPEC to three competing explanation uncertainty estimation methods. BayesSHAP and BayesLIME (Slack et al., 2021) are extensions of KernelSHAP and LIME, respectively, that fit Bayesian linear regression models to perturbed data samples. After fitting, 95% credible intervals can be estimated by sampling the posterior distribution of the feature attributions. CXPlain (Schwab & Karlen, 2019) trains a separate explanation model using a causal-based objective, and applies a bootstrap resampling approach to estimate explanation uncertainty. We report the 95% confidence interval from the set of bootstrapped explanations. Unless otherwise stated, we use the default settings in the provided implementation for all three of these methods.

## 5.2 UNCERTAINTY VISUALIZATION

In order to visualize explanation uncertainty, we train XGBoost binary classifiers using two selected features, which we take to be the black-box model. Both GPEC and competing methods are used to quantify the uncertainty for the feature attributions of the test samples: a grid of 10,000 samples over the data domain. Uncertainty estimates for the x-axis feature are plotted in Figure 4 (results for the y-axis variable are included in App. F.4), where darker values in the heatmap indicate higher uncertainty. GPEC-WEG, GPEC-RBF, and CXPlain require training samples for their amortized uncertainty estimates; these samples, also used as the reference distribution for BayesSHAP and BayesLIME, are plotted in red. The DB is plotted as the black line. We ablate the function approximation uncertainty component of GPEC-WEG and GPEC-RBF in order to evaluate the effects of their respective kernels. Comparing GPEC-WEG and GPEC-RBF shows that using the WEG kernel attributes higher uncertainty to samples near nonlinearities in the DB. In contrast, GPEC-RBF provides uncertainty estimates that relate only to the training samples; test sample uncertainty is proportional to distance from the training samples. The competing methods BayesSHAP, BayesLIME, and CXPlain results in relatively uniform uncertainty estimates over the test samples. CXPlain shows areas of higher uncertainty for Census, however the magnitude of these estimates are small. The uncertainty estimates produced by these competing methods are unable to capture the properties of the black-box model.

## 5.3 REGULARIZATION TEST

In this experiment we compare the average uncertainty of explanations as the model is increasingly regularized in order to assess the impact of restricting model complexity. For XGBoost models, we vary the parameter $\gamma$, which penalizes the number of leaves in the regression tree functions (Eq. 2 in Chen & Guestrin (2016)). For neural networks, we use two types of regularization. First, we add an $\ell_2$ penalty to the weights; this penalty increases as the parameter $\lambda$ increases. Second, we change the

| Dataset | Census | | | Online Shoppers | | | German Credit | | |
|---|---|---|---|---|---|---|---|---|---|
| Regularization | $\gamma$ | | | $\gamma$ | | | $\gamma$ | | |
| Magnitude | 0 | 5 | 10 | 0 | 5 | 10 | 0 | 5 | 10 |
| GPEC-WEG | 1.573 | 1.177 | 1.158 | 0.209 | 0.123 | 0.092 | 2.665 | 1.747 | 0.250 |
| GPEC-RBF | 0.493 | 0.472 | 0.466 | 1.740 | 1.731 | 1.734 | 0.153 | 0.081 | 0.073 |
| BayesSHAP | 0.037 | 0.037 | 0.037 | 0.031 | 0.031 | 0.031 | 0.019 | 0.019 | 0.018 |
| BayesLIME | 0.097 | 0.096 | 0.095 | 0.098 | 0.097 | 0.093 | 0.085 | 0.066 | 0.045 |
| CXPlain | 0.064 | 0.064 | 0.069 | 0.004 | 0.003 | 0.006 | $1.7e^{-4}$ | $1.1e^{-4}$ | $4.3e^{-4}$ |

| Dataset | MNIST | | | | | | Fashion MNIST | | | | | |
|---|---|---|---|---|---|---|---|---|---|---|---|---|
| Regularization | $\ell_2$ | | | Softplus $\beta$ | | | $\ell_2$ | | | Softplus $\beta$ | | |
| Magnitude | 0 | $1e^{-5}$ | $10e^{-5}$ | 1.0 | 0.5 | 0.25 | 0 | $1e^{-5}$ | $10e^{-5}$ | 1.0 | 0.5 | 0.25 |
| GPEC-WEG | 0.236 | 0.157 | 0.078 | 0.087 | 0.073 | 0.056 | 0.378 | 0.187 | 0.063 | 0.112 | 0.075 | 0.061 |
| GPEC-RBF | 0.439 | 0.301 | 0.232 | 0.226 | 0.232 | 0.236 | 1.01 | 0.424 | 0.247 | 0.302 | 0.261 | 0.250 |
| BayesSHAP | 0.025 | 0.016 | 0.008 | 0.013 | 0.011 | 0.010 | 0.030 | 0.014 | 0.007 | 0.018 | 0.010 | 0.009 |
| BayesLIME | 2.452 | 1.573 | 0.737 | 0.868 | 0.866 | 0.721 | 2.605 | 1.364 | 0.666 | 1.178 | 0.861 | 0.779 |
| CXPlain | $0.1e^{-5}$ | $5.0e^{-5}$ | $8.6e^{-5}$ | $5.3e^{-5}$ | $8.0e^{-5}$ | $5.4e^{-5}$ | $7.2e^{-5}$ | $4.8e^{-5}$ | $6.2e^{-5}$ | $9.0e^{-5}$ | $6.6e^{-5}$ | $9.6e^{-5}$ |

Table 1: Average explanation uncertainty estimates over all features for a given classifier with varying levels of regularization. Higher regularization (increasing left to right) generally results in smoother and simpler classifiers.

ReLU activation functions to Softplus; a smooth approximation of ReLU with smoothness inversely proportional to a parameter $\beta$, which we vary (Dombrowski et al., 2019). We observe in Table 1 that the average uncertainty estimate generated by GPEC-WEG decreases as regularization increases, suggesting that the uncertainty estimates reflect the overall complexity of the underlying black-box model. For the tabular datasets, the estimates for BayesSHAP, BayesLIME, and CXPlain stay relatively flat. Interestingly, the estimates from these methods decrease for the image datasets. We hypothesize that the regularization for the neural network model also increases overall stability of the explanations. GPEC can capture both the uncertainty from WEG kernel and also the estimated uncertainty from the function approximation for the explainer, which we demonstrate in Section 5.4.

## 5.4 COMBINING APPROXIMATION UNCERTAINTY

Using the WEG kernel with noisy explanation labels enables GPEC to incorporate uncertainty estimates from the explainer. Explainers such as BayesSHAP provide an estimate of uncertainty for their explanations, which can be used as fixed noise in the Gaussian likelihood for GPEC. For explainers with no such estimate, we can empirically resample explanations for the same test sample and take the explanation variance as an estimate of uncertainty. Here, we compare GPEC uncertainty results before and after applying explanation noise estimates. We combine GPEC with two different explainers: BayesSHAP and Shapley Sampling Values (SSV) (Strumbelj & Kononenko, 2013). The two selected explainers are different methods of approximating SHAP values, however the former has an inherent method for quantifying its uncertainty while the latter does not. In Figure 5 row (A) we plot the heatmap for uncertainty estimates using GPEC using only the WEG Kernel and no added noise from the explainers. In row (B) we add the additional uncertainty estimates from BayesSHAP and SSV. The difference in quantified uncertainty is visualized in the row (C). We observe that combining the uncertainty estimates from BayesSHAP and SSV in the GPEC Gaussian Process formulation increases the overall uncertainty estimate.

## 6 LIMITATIONS AND CONCLUSION

Generating uncertainty estimates for feature attribution explanations are essential for building reliable explanations. We introduce a novel GP-based approach that can be used with any black-box classifier and feature attribution method. GPEC generates uncertainty estimates for explanations that capture the complexity of the black-box model. Experiments show that capturing this uncertainty improves understanding of the explanations and the black-box model itself. Regarding limitations, GPEC relies on DB estimation methods which is an ongoing area of research. Due to the time complexity of DB estimation, this can result in a tradeoff between computation time and approximation accuracy or sample bias. However, the effects DB sampling time are minimized during inference as the DB only needs to be sampled during training. Additionally, in its current implementation GPEC is limited to black-box classifiers; we leave the extension to regression as future work.

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

## A   Societal Impacts

As machine learning models are increasingly relied upon in a diverse set of high-impact domains ranging from health-care to financial lending (Esteva et al., 2019; Kose et al., 2021; Doshi-Velez & Kim, 2017; Sheikh et al., 2020; Singh et al., 2021), it is crucial that users of these models can accurately interpret why predictions are made. An understanding of why a model is making a certain prediction is important for users to trust it – for instance a doctor may wish to know if a skin-cancer classifier's high test-set accuracy comes from the leveraging of truly diagnostic features, or a specific imaging device artifact. However, further spurred by the advent of deep learning's increasing popularity (Krizhevsky et al., 2017), many of the models deployed in these high-stakes fields are complex black box's; producing predictions which are non-trivial to explain the reasoning behind. The development of many methods for explaining black-box predictions has arisen from this situation (Ribeiro et al., 2016; Lundberg & Lee, 2017; Covert et al., 2020; Masoomi et al., 2021), but explanations may have varying quality and consistency. Before utilizing explanations in practice, it is essential that users know when, and when not, to trust them. Explanation uncertainty is one proxy for this notion of trust, in which more uncertain explanations may be deemed less trustworthy. In this work, we explore a new way to model explanation uncertainty, in terms of local decision-boundary complexity. In tandem with the careful consideration of domain experts, our methodology may be used to assist in determining when explanations are reliable. Our theoretical results provide new insights towards what explanation uncertainty entails, and open the door for future methods expounding upon our formulation.

## B   Background

### B.1   Related Works: Reliability of Explanations

While feature attribution methods have gained wide popularity, a number of issues relating to the reliability of such methods have been uncovered. Alvarez-Melis & Jaakkola (2018) investigate the notion of robustness and show that many feature attribution methods are sensitive to small changes in input. This has been further investigated in the adversarial setting for perturbation-based methods (Slack et al., 2020) and neural network-based methods. (Ghorbani et al., 2019). Kindermans et al. (2019) show that many feature attribution methods are affected by distribution transformations such as those common in preprocessing. The generated explanations can also be very sensitive to hyperparameter choice Bansal et al. (2020). A number of metrics have been proposed for evaluating explainer reliability, such as with respect to adversarial attack (Hsieh et al., 2021), local perturbations (Alvarez-Melis & Jaakkola, 2018; Visani et al., 2022), black-box smoothness (Khan et al., 2022), fidelity to the black-box model (Yeh et al., 2019), or combinations of these metrics (Bhatt et al., 2020).

### B.2   Gaussian Process Review

A single-output Gaussian Process represents a distribution over *functions* $f : \mathcal{X} \to \mathbb{R}$

$$f(x) \sim \mathcal{GP}(m(x), k(x, x')). \tag{10}$$

Here $m : \mathcal{X} \to \mathbb{R}$ and $k : (\mathcal{X}, \mathcal{X}) \to \mathbb{R}$ are the mean and kernel (or covariance) functions respectively, which are chosen *a priori* to encode the users assumptions about the data. The kernel function $k(x, x')$ reflects a notion of similarity between data points for which predictive distributions over $f(x), f(x')$ respect. The prior $m(x)$ – frequently considered to be less important – is commonly chosen to be the constant $m(x) = 0$.

Specifically, a GP is an infinite collection of R.V's $f(x)$, each indexed by an element $x \in \mathcal{X}$. Importantly, any finite sub-collection of these R.V's

$$f(X_{tr}) = (f(x_1) \ldots, f(x_n)) \in \mathbb{R}^d, \tag{11}$$

corresponding to some index set $X_{tr} = \{x_i\}_{i=1}^n \subset \mathcal{X}$, follows the multivariate normal (MVN) distribution, i.e.

$$f(X_{tr}) \sim \mathcal{N}(m(X_{tr}), K(X_{tr}, X_{tr})). \tag{12}$$

Here the mean vector $m(X_{tr}) = (m(x_1), \ldots, m(x_n)) \in \mathbb{R}^n$ represents the mean function applied on each $x \in X_{tr}$ and the covariance matrix $K \in \mathbb{R}^{n \times n}$, also known as the gram matrix, contains

each pairwise kernel-based similarity value $K_{ij} = k(x_i, x_j)$. Kernel function outputs correspond to dot products in potentially infinite dimensional expanded feature space, which allows for the encoding of nuanced notions of similarity; e.g. the exponential geodesic kernel referenced in this work (Feragen et al., 2015).

Making predictions with a GP is analogous to simply conditioning this normal distribution on our data. Considering a set of input,noise-free label pairs

$$\mathcal{D} = \{(x_i, f(x_i)\}_{i=1}^n \tag{13}$$

we may update our posterior over *any subset* of the R.V's $f(x)$ by considering the joint normal over the subset and $\mathcal{D}$ and conditioning on $\mathcal{D}$. For instance, when choosing a singleton index set $\{x_0\}$, the posterior over $f(x_0)|D$ is another normal distribution which may be written as[2]

$$f(x_0) \sim \mathcal{N}(\bar{f}(x_0), \mathbb{V}[f(x_0)]) \tag{14}$$

where

$$\bar{f}(x_0) = K(x_0, X_{tr})^T K(X_{tr}, X_{tr})^{-1} f(X_{tr}) \tag{15}$$

$$\mathbb{V}[f(x_0)] = k(x_0, x_0) - K(x_0, X_{tr})^T K(X_{tr}, X_{tr})^{-1} K(x_0, X_{tr}) \tag{16}$$

and $K(x_0, X_{tr}) \in \mathbb{R}^d$ is defined element-wise by $K(x_0, X_{tr})_i = k(x_0, x_i)$.

Now we may consider the situation where our labels are noisy:

$$\mathcal{D} = \{(x_i, y_i)\}_{i=1}^n, \ y_i = f(x_i) + \epsilon, \ \epsilon \sim \mathcal{N}(0, \sigma^2), \ \sigma^2 \in \mathbb{R}_+. \tag{17}$$

Here $y_i$ is equal to the quantity we wish to model, $f(x_i)$, with the addition of noise variable $\epsilon$. The conditional is still a MVN, but the mean and variance equations are slightly modified

$$\bar{f}(x_0) = K(x_0, X_{tr})^T (K(X_{tr}, X_{tr}) + \sigma^2 I)^{-1} Y \tag{18}$$

$$\mathbb{V}[f(x_0)] = k(x_0, x_0) - K(x_0, X_{tr})^T (K(X_{tr}, X_{tr}) + \sigma^2 I)^{-1} K(x_0, X_{tr}), \tag{19}$$

where $Y \in \mathbb{R}^n$ has elements $Y_i = y_i$.

Notice how the variance $\sigma^2 I$ is added to $K(X_{tr}, X_{tr})$ in the quadratic form in Eq. 19, resulting in smaller eigenvalues after matrix inversion. Since this quadratic form is subtracted, the decision to model labels as noisy increases the uncertainty (variance) of estimates the GP posterior provides. This agrees with the intuition that noisy labels should result in more uncertain predictions.

While GPs may also be defined over vector valued functions, in this work the independence of each output component is assumed, allowing for modeling with $c \geq 1$ independent GPs. For more details see Ch.2 of Rasmussen & Williams (2005), from which the notation and content of this section were inspired.

## C    PROOF OF THEOREMS AND MULTICLASS EXTENSION

### C.1    THEOREM 1: RELATION TO EXPONENTIAL GEODESIC KERNEL

$$k(x, y) = \int \int \exp[-\lambda d_{\text{geo}}(m, m')] q(m|x, \rho) q(m'|y, \rho) \ dm' dm$$

$$s.t. \quad q(m|x, \rho) \propto \exp[-\rho||x - m||_2^2] p(m)$$

Note that $\rho$ controls how to weight manifold samples close to $x, y$. We take $\lim_{\rho \to \infty}$:

$$\lim_{\rho \to \infty} q(m|x, \rho) q(m'|y, \rho) = \begin{cases} 1 & x = m \text{ and } y = m' \\ 0 & \text{Otherwise} \end{cases}$$

Therefore the function within the integral of $k(x, y)$ evaluates to zero at all points except $x = m$ and $y = m'$. Since $x, y \in \mathcal{M}$ we can evaluate the integral:

$$k(x, y) = \exp[-\lambda d_{\text{geo}}(x, y)]$$

---

[2] assuming prior $m(x) = 0$

## C.2 THEOREM 2: KERNEL SIMILARITY AND DECISION BOUNDARY COMPLEXITY

From definition 1, given any perturbation $\widetilde{\mathcal{P}}$ on $\mathcal{P}$, there must exist a compact subset $K_i \subset U_i$ s.t. $R|_{\mathcal{P}\setminus\text{int}(K_i)} = \text{id}|_{\mathcal{P}\setminus\text{int}(K_i)}$ and $R|_{\text{int}(K_i)} \neq \text{id}|_{\text{int}(K_i)}$. Furthermore there exists a linear homeomorphism between an open subset $\widetilde{U_i} \subseteq U_i$ with $\mathbb{R}^{d-1}$ which contains $K_i$.

We parametrize $K_i$ using a smooth function $g : \mathcal{T} \to K_i$ s.t. $g(t) \in \partial K_i \; \forall t \in \partial \mathcal{T}$.

We further define $g_\epsilon(t) = g(t) + \epsilon\eta(t)$, for some perturbation $\epsilon \in \mathbb{R}$ and a smooth function $\eta : \mathcal{T} \to \mathbb{R}^{d-1}$ We also restrict $\eta$ such that $\eta(t) = \mathbf{0} \; \forall t \in \partial\mathcal{T}$ and $\exists \; t_0 \in \mathcal{T}$ s.t. $\eta(t_0) \neq g(t_0)$. In other words, $\eta$ is a smooth function where $g_\epsilon(t) = g(t) \; \forall \epsilon > 0, \forall t \in \partial\mathcal{T}$, but is not identical to $g$ for all $t \in \mathcal{T}$. Using $g_\epsilon(t)$, we define the manifold $\mathcal{P}_\epsilon = \{g_\epsilon(t) : t \in \mathcal{T}\}$.

To complete the proof, we want to show that the kernel similarity between any two given points $x, y \in \mathbb{R}^d$ is lower when using the manifold $\mathcal{P}_\epsilon$ for $\epsilon > 0$ as opposed to the manifold $\mathcal{P}_0$. We therefore want to compare the two respective kernels $k_\epsilon(x, y)$ and $k_0(x, y)$. Note that in this proof we consider the local effects of $\mathcal{P}$ on the kernel similarity through $\mathcal{P}_0$ and $\mathcal{P}_\epsilon$ exclusively, ignoring the manifold $\mathcal{P} \setminus U_0$ Using Euler-Lagrange, we can calculate a lower bound for $d_{\text{geo}}(g_\epsilon(t)), g_\epsilon(t')))$. In particular, for any $t, t' \in \mathcal{T}$, $d_{\text{geo}}(g_\epsilon(t)), g_\epsilon(t'))) \geq d_{\text{geo}}(g_0(t), g_0(t'))$.

$$d_{\text{geo}}(g_\epsilon(t), g_\epsilon(t')) \geq d_{\text{geo}}(g_0(t), g_0(t')) \tag{20}$$

$$\exp[-\lambda d_{\text{geo}}(g_\epsilon(t), g_\epsilon(t'))] \leq \exp[-\lambda d_{\text{geo}}(g_0(t), g_0(t'))] \tag{21}$$

$$\int_\mathcal{T} \int_\mathcal{T} \exp[-\lambda d_{\text{geo}}(g_\epsilon(t), g_\epsilon(t'))] \, \mathrm{d}t\mathrm{d}t' \leq \int_\mathcal{T} \int_\mathcal{T} \exp[-\lambda d_{\text{geo}}(g_0(t), g_0(t'))] \, \mathrm{d}t\mathrm{d}t' \tag{22}$$

Note that in Eq. 22 we are integrating over all possible values of $t, t'$, therefore the inequality is tight iff $g_\epsilon(t) = g_0(t) \; \forall t \in \mathcal{T}$; i.e. $\epsilon = 0$ (see proof in C.2.1). The case of $\epsilon = 0$ is trivial; we instead assume $\epsilon > 0$, in which case we can establish the following strict inequality:

$$\int_\mathcal{T} \int_\mathcal{T} \exp[-\lambda d_{\text{geo}}(g_\epsilon(t), g_\epsilon(t'))] \, \mathrm{d}t\mathrm{d}t' < \int_\mathcal{T} \int_\mathcal{T} \exp[-\lambda d_{\text{geo}}(g_0(t), g_0(t'))] \, \mathrm{d}t\mathrm{d}t' \tag{23}$$

Define uniform random variables $T, T'$ over the domain of $g$, i.e. $T, T' \sim \mathcal{U}_\mathcal{T}$. Then we have:

$$\mathbb{E}_{T,T'\sim\mathcal{U}_{[0,1]}}[\exp[-\lambda d_{\text{geo}}(g_\epsilon(T), g_\epsilon(T'))]] < \mathbb{E}_{T,T'\sim\mathcal{U}_{[0,1]}}[\exp[-\lambda d_{\text{geo}}(g_0(T), g_0(T'))]] \tag{24}$$

$$\mathbb{E}_{M,M'\sim p_\epsilon(M)}[\exp[-\lambda d_{\text{geo}}(M, M')]] < \mathbb{E}_{M,M'\sim p_0(M)}[\exp[-\lambda d_{\text{geo}}(M, M')]] \tag{25}$$

We define the random variable $M = g_\epsilon(T)$ with distribution $p_\epsilon(M)$. The distribution $p_\epsilon(M)$ represents the uniform distribution $\mathcal{U}_\mathcal{T}$ mapped to the manifold $\mathcal{P}_\epsilon$ using $g_\epsilon(T)$. The step from Eq. 24 to Eq. 25 uses a property of distribution transformations (Eq. 2.2.5 in Casella & Berger (2001)).

Next, compare either side of Eq. 25 to our kernel formulation shown below in Eq. 26. The kernel $k_\epsilon(x, y|\rho, \lambda)$ takes an expected value over $q_\epsilon(M|x, \rho)$ and $q_\epsilon(M'|y, \rho)$, which are equivalent to $p_\epsilon(M)$ and $p_\epsilon(M')$ weighted with respect to $x, y$, and a hyperparameter $\rho \geq 0$.

$$k_\epsilon(x, y|\rho, \lambda) = \mathbb{E}_{M\sim q_\epsilon(M|x,\rho), M'\sim q_\epsilon(M'|y,\rho)}[\exp[-\lambda d_{\text{geo}}(M, M')]] \tag{26}$$
$$s.t. \quad q_\epsilon(M|x, \rho) \propto \exp[-\rho\|x - M\|_2^2]p_\epsilon(M)$$
$$s.t. \quad q_\epsilon(M'|y, \rho) \propto \exp[-\rho\|y - M'\|_2^2]p_\epsilon(M')$$

Note that when $\rho$ is set to zero, $q(M|x, 0) = p(M)$ and $q(M'|y, 0) = p(M')$. Therefore Eq. 25 is equivalent to the inequality $k_\epsilon(x, y|0, \lambda) < k_0(x, y|0, \lambda)$.

We next want to prove that the inequality $k_\epsilon(x, y|\rho, \lambda) < k_0(x, y|\rho, \lambda)$ also holds for non-zero values of $\rho$. For convenience, define

$$f(\rho) = k_0(x, y|\rho, \lambda) - k_\epsilon(x, y|\rho, \lambda) \tag{27}$$

Under this definition, we want to prove there exists $\rho_0 > 0$ such that $f(\rho) > 0 \; \forall \rho < \rho_0$. From Eq. 25, we established that $f(0) > 0$. Assume that

$$\lim_{\rho \to 0} f(\rho) = c \tag{28}$$

It therefore follows that $c > 0$. In addition, note that $f(\rho)$ is continuous with respect to $\rho$ (see proof in section C.2.3). Therefore for any $\epsilon > 0$ there exists $\delta > 0$ s.t. $\rho < \delta$ implies $|f(\rho) - c| < \epsilon$.

We choose $\epsilon = c$ and the define the corresponding $\delta$ to be $\rho_0$. Therefore:

$$\rho < \rho_0 \Rightarrow |f(\rho) - c| < c \tag{29}$$

$$\rho < \rho_0 \Rightarrow 0 < f(\rho) < 2c \tag{30}$$

Since this result holds for any $i$, it follows that the piecewise linear manifold $\mathcal{P}$ is a local minimum under any perturbation along a specific chart or combination of charts with respect to the kernel similarity $k(x, y) \; \forall x, y \in \mathbb{R}^d$.

### C.2.1 PROOF: EQ. 23

We want to prove:

$$\int_\mathcal{T} \int_\mathcal{T} \exp[-\lambda d_{\text{geo}}(g_\epsilon(t), g_\epsilon(t'))] \, \mathrm{d}t \mathrm{d}t' = \int_\mathcal{T} \int_\mathcal{T} \exp[-\lambda d_{\text{geo}}(g_0(t), g_0(t'))] \, \mathrm{d}t \mathrm{d}t' \tag{31}$$
$$\Rightarrow g_\epsilon(t) = g_0(t) \quad \forall t \in \mathcal{T}$$

Consider the LHS of Eq. 31:

$$\int_\mathcal{T} \int_\mathcal{T} \exp[-\lambda d_{\text{geo}}(g_\epsilon(t), g_\epsilon(t'))] \, \mathrm{d}t \mathrm{d}t' = \int_\mathcal{T} \int_\mathcal{T} \exp[-\lambda d_{\text{geo}}(g_0(t), g_0(t'))] \, \mathrm{d}t \mathrm{d}t' \tag{32}$$

$$\int_\mathcal{T} \int_\mathcal{T} \underbrace{\exp[-\lambda d_{\text{geo}}(g_0(t), g_0(t'))] - \exp[-\lambda d_{\text{geo}}(g_\epsilon(t), g_\epsilon(t'))]}_{h(t, t')} \, \mathrm{d}t \mathrm{d}t' = 0 \tag{33}$$

Define $h(t, t')$ as the function inside the integrals in Eq. 33. From Eq. 21, $h(t, t') \geq 0 \; \forall t, t' \in \mathcal{T}$. Since $h$ is continuous (see proof in C.2.2) and $\int_\mathcal{T} \int_\mathcal{T} h(t, t') \mathrm{d}t \mathrm{d}t' = 0$, it follows that $h(t, t') = 0$ $\forall t, t' \in \mathcal{T}$ (Ch.6 Rudin (1976)).

It therefore follows that:

$$\exp[-\lambda d_{\text{geo}}(g_0(t), g_0(t'))] = \exp[-\lambda d_{\text{geo}}(g_\epsilon(t), g_\epsilon(t'))] \quad \forall t, t' \in \mathcal{T} \tag{34}$$

From the definition of $\eta(t)$ in $g_\epsilon(t) = g(t) + \epsilon\eta(t)$, there must exist $t \in \mathcal{T}$ s.t. $\eta(t) \neq 0$. Therefore $\epsilon$ must be zero for Eq. 34 to hold. It follows that $g_\epsilon(t) = g_0(t) \; \forall t \in \mathcal{T}$.

### C.2.2 PROOF: CONTINUITY OF $h(t, t')$

We prove that $h(t, t')$ is continuous with respect to $t, t'$. First note that by definition, $g_\epsilon(t)$ is a continuous parametrization of the manifold $\mathcal{P}_\epsilon$. From Burago et al. (2001), it follows that for any two points $g_\epsilon(t), g_\epsilon(t') \in \mathcal{P}_\epsilon$, $d_{\text{geo}}(g_\epsilon(t), g_\epsilon(t'))$ is continuous. Since the exponential functional preserves continuity and the sum of continuous functions are also continuous, it follows that $h(t, t')$ is continuous.

### C.2.3 PROOF: CONTINUITY OF $k(x, y)$ WITH RESPECT TO $\rho$

We prove that $k(x, y)$ is continuous with respect to $\rho$.

$$k(x, y) = \int \int \exp[-\lambda d_{\text{geo}}(m, m')] \, q(m|x, \rho) \, q(m'|y, \rho) \, \mathrm{d}m \mathrm{d}m' \tag{35}$$

$$= \frac{1}{Z_m(\rho) Z_{m'}(\rho)} \int \int \mathcal{A} \underbrace{\exp[-\rho(||x - m||_2^2 + ||y - m'||_2^2)]}_{Z(\rho)} \, \mathrm{d}m \mathrm{d}m' \tag{36}$$

$$s.t. \quad Z_m(\rho) = \int \exp[-\rho ||x - m||_2^2] p(m) \, \mathrm{d}m$$

$$Z_{m'}(\rho) = \int \exp[-\rho ||y - m'||_2^2] p(m') \, \mathrm{d}m'$$

$$\mathcal{A} = \exp[-\lambda d_{\text{geo}}(m, m')] p(m) p(m')$$

Define $h(\rho) = \rho \mathcal{B}$, where $\mathcal{B}$ is a constant. Consider $h(\rho) - h(\rho_0)$, where $\rho_0$ is a fixed positive constant:

$$|h(\rho) - h(\rho_0)| = |\rho \mathcal{B} - \rho_0 \mathcal{B}| \tag{37}$$

$$= |(\rho - \rho_0)\mathcal{B}| < \delta |\mathcal{B}| \tag{38}$$

It follows that $\forall \, \epsilon > 0, \exists \, \delta = \frac{\epsilon}{|\mathcal{B}|} > 0$ such that $|\rho - \rho_0| < \delta \Rightarrow |h(\rho) - h(\rho_0)| < \epsilon$. Therefore $h$ is continuous for all $\rho \in \mathbb{R}^+$.

We set $\mathcal{B}$ to be $||x - m||_2^2$, $||y - m'||_2^2$, and $||x - m||_2^2 + ||y - m'||_2^2$, which shows that $Z_m(\rho)$, $Z_{m'}(\rho)$, and $Z(\rho)$ are also continuous, respectively. It then follows that the entirety of Eq. 36 is continuous.

### C.3 EXTENDING TO MULTICLASS CLASSIFIERS

In the multiclass case we define a black-box prediction model $F : \mathcal{X} \to \mathbb{R}^c$. We consider the one-vs-all DB for every class $y \in \mathcal{Y} = \{1, \dots, c\}$, defined as $\mathcal{M}_y = \{x \in \mathbb{R}^d : F_y(x) = \max_{i \in \mathcal{Y}} F_i(x) = \max_{j \neq y \in \mathcal{Y}} F_j(x)\}$, where $F_k$ indicates the model output for class $k$. We then apply the GPEC framework separately to each class using the respective DB. The uncertainty estimate of the GP model would be of dimension $d \times s$.

## D IMPLEMENTATION DETAILS

### D.1 ALGORITHM

The GPEC training algorithm is outlined in Alg. 1. GPEC is parametrized using a multi-output Gaussian Process Regression model using the explanations as labels. Once the explanations $L$, explanation uncertainty $U$, and WEG kernel matrix $K$ are generated from Alg. 1, we can directly use these values to update the GP posterior and calculate the prediction variance for new test samples (Eq. 19).

## D.2 ADVERSARIAL SAMPLE FILTERING MULTI-CLASS MODELS

We elect to sample from multi-class neural network decision boundaries by running binary search on train-point adversarial example pairs. Specifically, given a test-point $x_0 \in \mathbb{R}^d$ and model prediction $y = \text{argmax}_{k \in \mathcal{Y}} F(x_0)$, decision boundary points may be generated by the following procedure:

First, for each class $v \in \mathcal{Y}$ a set of $M_v$ points is randomly sampled from the set of train points on which the model predicts class $v$:

$$\mathcal{X}_v \subseteq \{x : \text{argmax}_{k \in \mathcal{Y}} F(x) = v, \ x \in \mathcal{X}_{tr}\}, \ |\mathcal{X}_v| = M_v \qquad (39)$$

$\forall v \in \mathcal{Y}$. An untargeted adversarial attack using some $l_p$ norm and radius $\epsilon$ is run on each point in $\mathcal{X}_y$, the set of points with the same class prediction as $x_0$. Each attack output $Attack_{un}(x, \epsilon) \in \mathbb{R}^d$ is paired with its corresponding input, resulting in the set

$$\mathcal{X}_{y'} = \{(x, Attack_{Un}(x, \epsilon)) : x \in \mathcal{X}_y\}, \qquad (40)$$

where for an element $(a, b) \in \mathcal{X}_{y'}$ we have $\text{argmax}_{k \in \mathcal{Y}} F(a) = y$, $\text{argmax}_{k \in \mathcal{Y}} F(b) = v \neq y$, where $v$ is an unspecified class.

Likewise a targeted adversarial attack, with target class $y$, is run on each point in each of the sets of points that are not predicted as class $y$. Each attack output $Attack_y(x, \epsilon) \in \mathbb{R}^d$ may be paired with its input $x$ resulting in sets

$$\mathcal{X}_{v'} = \{(x, Attack_y(x, \epsilon)) : x \in \mathcal{X}_v\} \qquad (41)$$

$\forall v \neq y \in \mathcal{Y}$. Here, for an element $(a, b) \in \mathcal{X}_{v'}$ we have $\text{argmax}_{k \in \mathcal{Y}} F(a) = v$, $\text{argmax}_{k \in \mathcal{Y}} F(b) = y$.

Thus, we have generated a diverse set of $\sum_{v \in \mathcal{Y}} M_v$ pairs of points that lie on opposite sides of the decision boundary for class $y$. The segment between any pair from a given set $\mathcal{X}_{v'}$ $v \neq y$ will necessarily contain a point on the class $v$ v.s. class $y$ decision boundary. Likewise, in the interest of further diversity, segments between any pair from the set $\mathcal{X}_{y'}$ will contain a point on the class $v$ v.s. class $y$ decision boundary, where $v \neq y \in \mathcal{Y}$ is unspecified. A binary search may be run on each pair to find the boundary point in the middle.

In practice the entire procedure may be amortized for each class, and ran for all classes as a single post-processing step immediately after training. This results in a dictionary of boundary points which may be efficiently queried on demand via the model predicted class of any given test point.

Each adversarial attack is attempted multiple times, once using each radius value $\epsilon$ in the list: $[0.0, 2e^{-4}, 5e^{-4}, 8e^{-4}, 1e^{-3}, 1e^{-3}, 1.5e^{-3}, 2e^{-3}, 3e^{-3}, 1e^{-2}, 1e^{-1}, 3e^{-1}, 5e^{-1}, 1.0]$. For a given input, the output of the successful attack with smallest $\epsilon$ is used. If no attack is successful at any radius, the input is discarded from further consideration.

In this work the Foolbox Rauber et al. (2017; 2020) implementation of the Projected Gradient Descent (PGD) Madry et al. (2018) attack with the $l_\infty$ norm was used for both targeted and untargeted attacks. The $M_c$ values used for the relevant datasets are indicated below in Appendix E.1.

## E EXPERIMENT SETUP

### E.1 DATASETS AND MODELS

**Census.** The UCI Census dataset consists of 32,561 samples from the 1994 census dataset. Each sample is a single person's response to the census questionaire. An XGBoost model is trained using the 12 features to predict whether the individual has income $\geq$ \$50k.

**Online Shopper.** The UCI Online Shoppers dataset consists of clickstream data from 12,330 web sessions. Each session is generated from a different individual and specifies whether a revenue-generating transaction takes place. There are 17 other features including device information, types of pages accessed during the session, and date information. An XGBoost model is trained to predict whether a purchase occurs.

**German Credit.** The German Credit dataset consists of 1,000 samples; each sample represents an individual who takes credit from a bank. The classification task is to predict whether an individual is

|  | Census | Online Shoppers | German Credit | MNIST | f-MNIST |
|---|---|---|---|---|---|
| GPEC-WEG | 0.11 | 0.37 | 0.07 | 12.90 | 18.15 |
| GPEC-RBF | 0.00 | 0.00 | 0.02 | 8.95 | 7.41 |
| CXPlain | 0.05 | 0.06 | 0.04 | 9.76 | 18.18 |
| BayesSHAP | 140.40 | 54.56 | 4.86 | 42,467 | 42,361 |
| BayesLIME | 91.29 | 54.60 | 4.83 | 41,832 | 41,992 |

Table 2: Execution time comparison for estimating the uncertainty for all features for 100 samples (in seconds). For MNIST and f-MNIST datasets, results represent execution time for calculating uncertainty estimates with respect to all ten classes. For GPEC-WEG, GPEC-RBF, and CXPlain methods, the results show inference times.

considered a good or bad risk. Features include demographic information, credit history, and information about existing loans. Categorical features are converted using a one-hot encoding, resulting in 24 total features.

**MNIST.** The MNIST dataset (LeCun & Cortes, 2010) consists of 70k grayscale images of dimension 28x28. Each image has a single handwritten numeral, from 0-9. A fully connected network with layer sizes 784-700-400-200-100-10 and ReLU activation functions was trained and validated on on $50,000$ and $10,000$ image label pairs, respectively. Training lasted for 30 epochs with initial learning rate of 2 and a learning rate decay of $\gamma = 0.5$ when training loss is plateaued. During adversarial example generation we used $M_y = 500$ and $M_c = 50 \ \forall c \neq y$.

**Fashion MNIST** The Fashion MNIST dataset (Xiao et al., 2017) contains 70k grayscale images of dimension 28x28. There are 10 classes, each indicating a different article of clothing. We train a MLP model with the same architecture used for the MNIST dataset, however we increase training to 100 epochs and increase the initial learning rate to 3. During adversarial example generation we used $M_y = 500$ and $M_c = 50 \ \forall c \neq y$.

### E.2 COMPETITOR IMPLEMENTATION DETAILS

**BayesLIME and BayesSHAP.** Slack et al. (2021) extend the methods LIME and KernelSHAP to use a Bayesian Framework. BayesLIME and BayesSHAP are fit using Bayesian linear regression models on perturbed outputs of the black-box model. The posterior distribution of the model weights are taken as the feature attributions instead of the frequentist estimate that characterizes LIME and KernelSHAP. We take the expected value of the posterior distribution as the point estimate for feature attributions, and the 95% credible interval as the estimate of uncertainty. To implement BayesLIME and BayesSHAP we use the public implementation[3]. We set the number of samples to 200, disable discretization for continuous variables, and calculate the explanations over all features. Otherwise, we use the default parameters for the implementation.

**CXPlain.** Schwab & Karlen (2019) introduces the explanation method CXPlain, which trains a surrogate explanation model based on a causal loss function. After training the surrogate model, the authors propose using a bootstrap resampling technique to estimate the variance of the predictions. In our experiments we implement the publicly available code[4]. We use the default parameters, which include using a 2-layer UNet model Ronneberger et al. (2015) for the image datasets and a 2-layer MLP model for the tabular datasets. We take a 95% confidence interval from the bootstrapped results as the estimate of uncertainty.

## F ADDITIONAL RESULTS

### F.1 EXECUTION TIME COMPARISON

In Table 2 we include an execution time comparison between the methods implemented in this paper. Results are averaged over 100 test samples. For MNIST and f-MNIST datasets, results evaluate the time to calculate uncertainty estimates with respect to all classes. All experiments were run on an internal cluster using AMD EPYC 7302 16-Core processors. We observe from the results that

---

[3]https://github.com/dylan-slack/Modeling-Uncertainty-Local-Explainability
[4]https://github.com/d909b/cxplain

the methods that amortization methods (GPEC-WEG, GPEC-RBF, CXPlain) are significantly faster than perturbation methods BayesLIME and BayesSHAP.

## F.2 Toy Example: Evaluating GPEC on a Linear Classifier

In order to understand how the GPEC-WEG uncertainty estimate behaves for linear models, we use a toy example shown in Fig. F.2. In the top row, we visualize the GPEC-WEG uncertainty; we see that the uncertainty estimate is a small, constant value for the linear model (left) whereas uncertainty increases for the nonlinear model (right). Intuitively, GPEC derives its uncertainty estimate by evaluating the distribution of explanations with respect to the black-box model. Since it is generally infeasible to evaluate the space of all possible explanations, GPEC will typically estimate a "baseline" amount of uncertainty for every explanation, which depends on the sampling of the training distribution and is minimized for a linear model.

In the bottom two figures we visualize the magnitude of the gradient for the two respective models, which is a noiseless estimate of feature importance. We see that even using this deterministic feature importance method, the estimates can fail to be robust due to the nonlinearity, i.e. nearby samples within the same class can have very different explanations.

## F.3 Visualizing effects of Explainer Uncertainty in GPEC Estimate

In section 5.4 we evaluate GPEC's ability to combine uncertainty from the black-box decision boundary and the uncertainty estimate from BayesSHAP and SSV explainers. In Figure 7 we extend this experiment to evaluate how well GPEC can capture the explainer uncertainty. We calculate the combined GPEC+explainer estimate using different numbers of approximation samples.

Both BayesSHAP and SSV depend on sampling to generate their explanations; having fewer samples increases the variance of their estimates. As we decrease the number of samples from 200 (Row A) to 5 (Row B) we would expect that the explainer uncertainty, and consequently the combined GPEC uncertainty, would increase. We see in Row $\Delta$ that the results follow our intuition; uncertainty increases for most of the plotted test points and uncertainty does not decrease for any points.

## F.4 Additional Results for Uncertainty Visualization Experiment

**Results for Y-Axis Feature** In Figure 8 we visualize the estimated explanation uncertainty as a heatmap for a grid of explanations. The generated plots only visualize the uncertainty for the feature on the x-axis. Due to space constraints, we list the results for the y-axis feature in the appendix, in Figure 8. We can see that the results are in line with those from the x-axis figure.

**Black-box model output** For reference, in Figure 9 we plot the probability output of the XGBoost models used in the visualization experiment (Figure 4).

## F.5 Sensitivity Analysis of WEG Kernel Parameters

The WEG kernel formulation uses two parameters, $\rho$ and $\lambda$. The parameter $\rho$ controls the weighting between each datapoint and the manifold samples. As $\rho$ increases, the WEG kernel places more weight on manifold samples close in $\ell_2$ distance to the given datapoint. The parameter $\lambda$ acts as a bandwidth parameter for the exponential geodesic kernel. Increasing $\lambda$ increases the effect of the geodesic distance along the manifold. Therefore decision boundaries with higher complexity will have an increased effect on the WEG kernel similarity. In Figures 10, 11, and 12 we plot heatmaps for various combinations of $\rho$ and $\lambda$ parameters to evaluate the change in the uncertainty estimate. The black line is the decision boundary and the red points are the samples used for training GPEC. Please note that the heatmap scales are not necessarily the same for each plot.

---

**Algorithm 1** GPEC Model Training

---

**Input :** GPEC Training Samples $X \in \mathbb{R}^{M \times d}$. Explainer $E$. Hyperparameters $P$ (# DB Samples), $J$ (*optional*: # Samples for Functional Approximation Uncertainty estimate)
**Output :** Explanations $L \in \mathbb{R}^{M \times S}$, Explanation Uncertainty $U \in \mathbb{R}^{M \times S}$, WEG Kernel $K \in [0, 1]^{M \times M}$

\\ Calculate Function Approximation Uncertainty
Initialize $L \in \mathbb{R}^{M \times S}$, $U \in \mathbb{R}^{M \times S}$
**if** *Explainer returns uncertainty estimate* **then**
    **for** *i = 1,2,...M* **do**
        $L_{i,:}, U_{i,:} \leftarrow E(X_{i,:})$       \\ Get explanations ($L_{i,:}$) and explanation uncertainty ($U_{i,:}$) from Explainer
    **end**
**end**
**else if** *Explainer is stochastic* **and** $J > 1$ **then**
    **for** *i = 1,2,...M* **do**
        Initialize $Q \in \mathbb{R}^{J \times S}$
        **for** *j = 1,2,...J* **do**
           $Q_{j,:} \leftarrow E(X_{i,:})$       \\ Draw stochastic explanations for same data sample
        **end**

        $L_{i,:} \leftarrow \dfrac{1}{J} \sum_{j=1}^{J} Q_{j,:}$

        $U_{i,:} \leftarrow \dfrac{1}{J} \sum_{j=1}^{J} (Q_{j,:} - L_{i,:})^2$       \\ Empirical uncertainty estimate (Eq. 2)

    **end**
**end**
**else if** *Explainer is deterministic* **then**
    **for** *i = 1,2,...M* **do**
        $L_{i,:} \leftarrow E(X_{i,:})$
    **end**
    $U \leftarrow \mathbf{0}$       \\ Set functional approximation uncertainty to zero
**end**

\\ Calculate EG Kernel Matrix
$B \leftarrow$ *Decision_Boundary_Sampler*$(F, P)$       \\ Draw $P$ DB samples of dimension $d$. $B \in \mathbb{R}^{P \times d}$
Initialize $G \in [0, 1]^{P \times P}$
**for** *i = 1,2,...P* **do**
    **for** *j = 1,2,...P* **do**
        $G_{i,j} \leftarrow \exp(-\lambda d_{geo}(B_{i,:}, B_{j,:}))$       \\ Eq. 4
    **end**
**end**

\\ Calculate WEG Kernel Matrix
Initialize $W \in [0, 1]^{M \times P}$       \\ Initialize weighting matrix
**for** *i = 1,2,...M* **do**
    **for** *j = 1,2,...P* **do**
        $W_{i,j} \leftarrow \exp(-\rho ||X_{i,:} - B_{j,:}||_2^2)$
    **end**
    $W_{i,:} \leftarrow \dfrac{W_{i,:}}{\sum_{j=1}^{P} W_{i,j}}$       \\ Normalize weighting distribution (Eq. 9)
**end**
$K \leftarrow WGW^{\mathsf{T}}$       \\ Apply weighting to EG Kernel Matrix

---

Return $L, U, K$

---

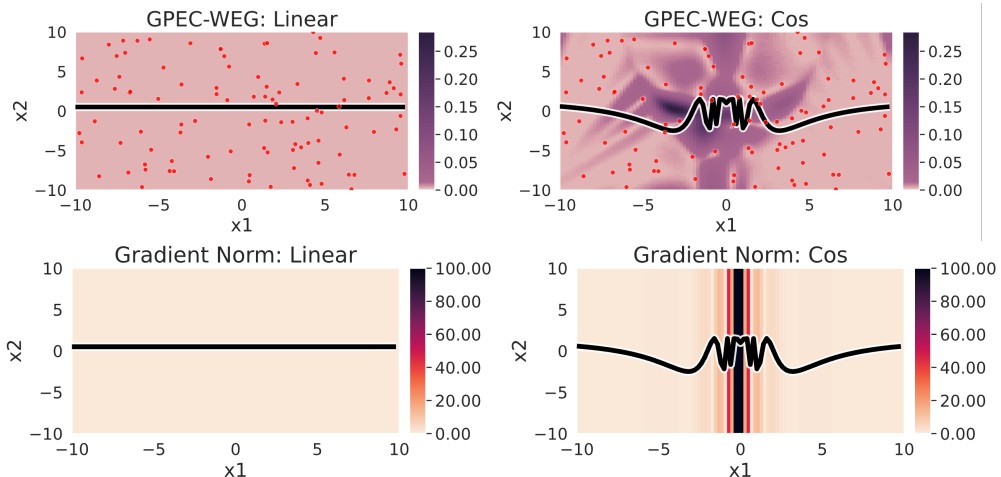

Figure 6: Comparison of a GPEC uncertainty estimates on linear and nonlinear toy models: $f_{\text{linear}}(x_1, x_2) = x_2$ and $f_{\cos}(x_1, x_2) = 2\cos(\frac{10}{x_1}) - x_2$. TOP: Uncertainty estimate from GPEC. Applying GPEC on a linear model results in a small, relatively constant variance for the test explanations. As the DB becomes more complex, as in $f_{\cos}$, the uncertainty estimate increases around the nonlinearities in the DB. BOTTOM: Gradient norm, which is an estimate of feature importance. The feature importance estimate becomes unstable (i.e. nearby samples of the same class have can have very different explanations) due to the nonlinearity of $f_{\cos}$.

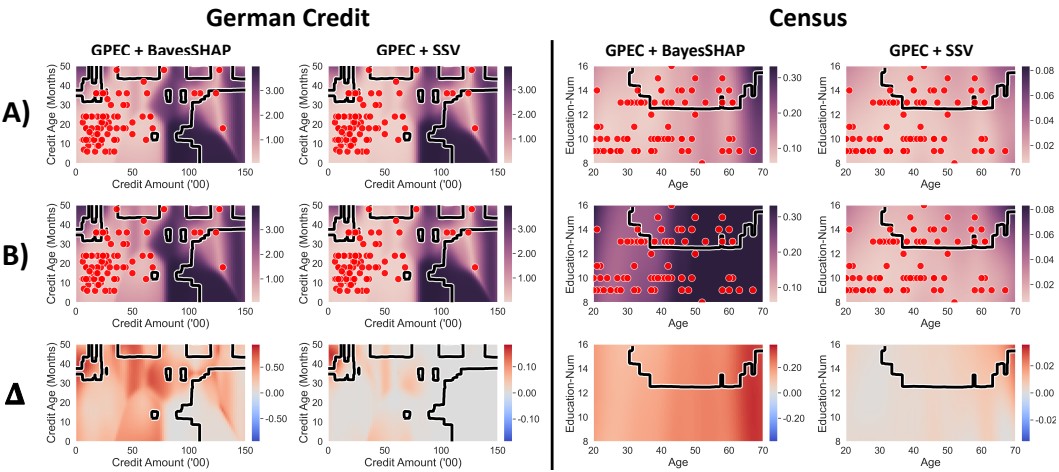

Figure 7: Comparison of the change in quantified uncertainty of explanations as we change the number of samples for BayesSHAP and SSV. Row (A) visualizes the combined uncertainty estimate using GPEC and either BayesSHAP or SSV, using 200 samples for approximating the BayesSHAP / SSV explanation. In Row (B) we decrease the number of samples to 5 and recalculate the estimated uncertainty. Row ($\Delta$) represents the change in uncertainty estimate between (A) and (B). We see that the average uncertainty changes as we decrease the number of samples, which indicates that GPEC is able to capture the uncertainty arising from BayesSHAP / SSV approximation.

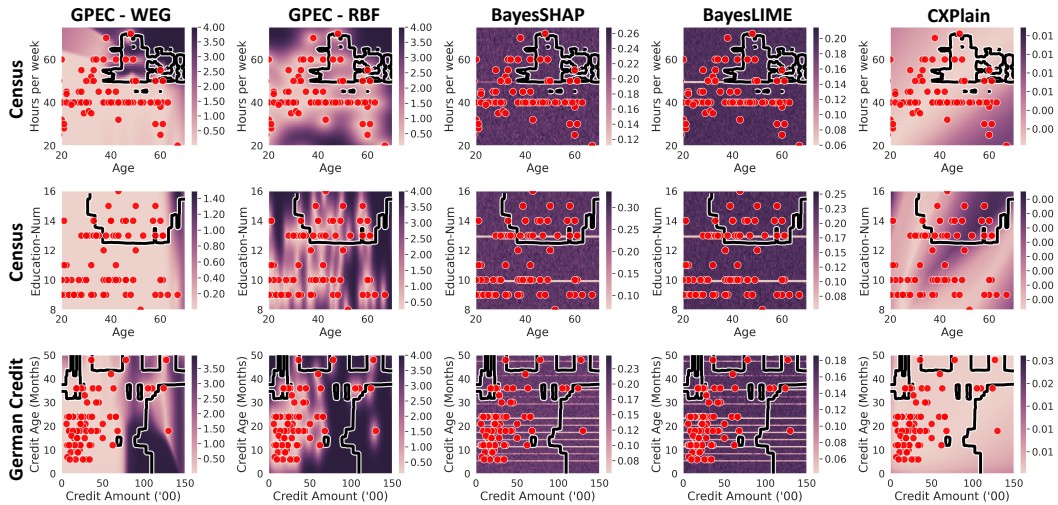

Figure 8: Complement to Figure 4. Visualization of estimated explanation uncertainty where the heatmap represents level of uncertainty for the feature on the **y-axis**.

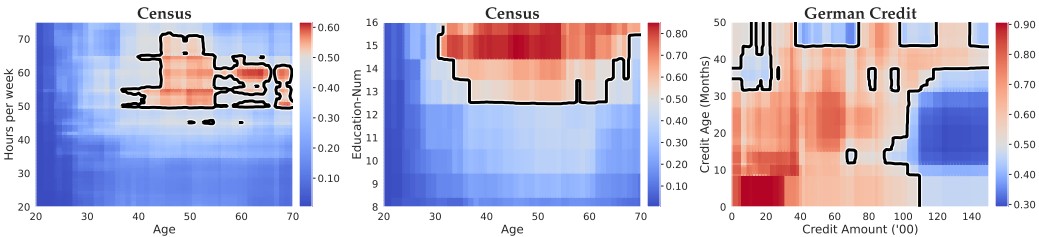

Figure 9: Output of binary classifiers used in the experiments. The color indicates the predicted probability of each point. Points with probability $\geq 0.5$ are classified as class 1, and points with probability $< 0.5$ are classified as class 0. The decision boundary, which is the set of points $\{m : m \in \mathbb{R}^2, f(m) = 0.5\}$ where $f$ is the classifier, is represented by the black line.

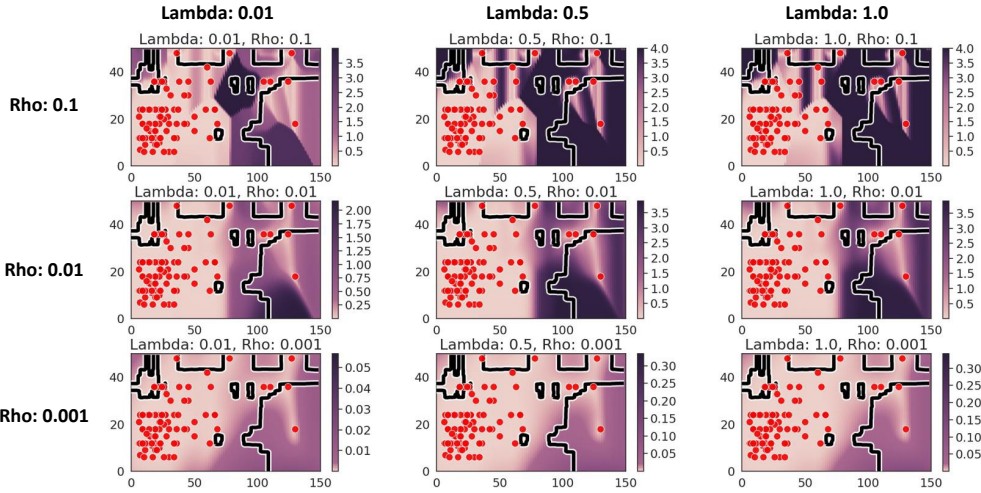

Figure 10: Hyperparameter sensitivity analysis for the German Credit Dataset. Heatmap of estimated uncertainty for the x-axis variable under different $\rho$ and $\lambda$ parameter choices.

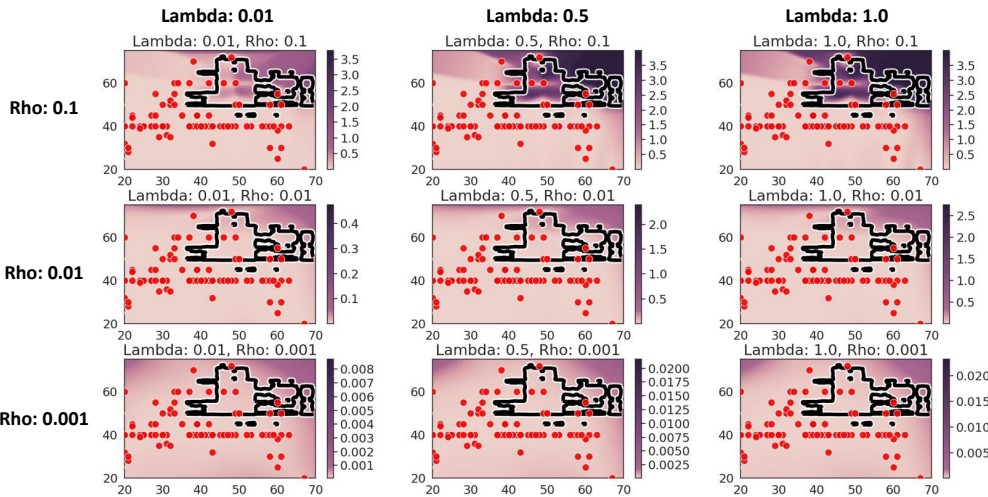

Figure 11: Hyperparameter sensitivity analysis for the Census Dataset. Heatmap of estimated uncertainty for the x-axis variable under different $\rho$ and $\lambda$ parameter choices.

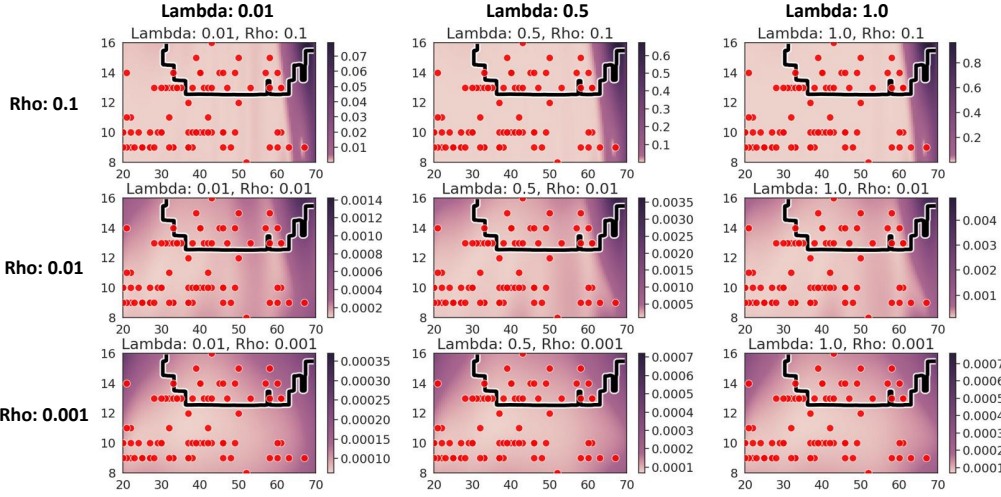

Figure 12: Hyperparameter sensitivity analysis for the Census Dataset. Heatmap of estimated uncertainty for the x-axis variable under different $\rho$ and $\lambda$ parameter choices.

