# OpenReview forum: "Explanation Uncertainty with Decision Boundary Awareness"
_ICLR.cc/2023/Conference — Submitted to ICLR 2023_

### Official Review · Reviewer_dnzT · 2022-10-23

**Confidence:** 3
**Correctness:** 3
**Technical Novelty And Significance:** 3
**Empirical Novelty And Significance:** 1
**Recommendation:** 3

**Clarity, Quality, Novelty And Reproducibility:**

Clarity: This paper is mostly clear and easy to read. The discussion around Theorem 2 is difficult to understand (at least to me), and some larger context regarding why this theorem is interesting would help.

Quality: The technical quality in this paper is moderate. While the method itself seems mostly fine (barring an issue discussed in the weaknesses section), the paper has poor experimental results which reduces its overall quality.

Novelty: The methods proposed in this paper are novel to the best of my knowledge and add to the literature on estimating uncertainty of explanation methods.

Reproducibility: The paper scores low on reproducibility, as the paper does not provide either the source code to reproduce experiments, or a succinct description of the exact algorithm used. In particular, the precise algorithm to compute the geodesic distances or sample decision boundary points are not given explicitly.

**Strength And Weaknesses:**

**Strengths**:

+ The paper tackles an important problem of estimating the uncertainty of explanations, which can be used to (potentially) judge whether or not trust the underlying explanation.

+ The proposed WEG kernel is a nice definition of the geodesic distance between two data points with respect to an underlying decision boundary. Arguments made to show that it constitutes a well-defined kernel, and that it is related to the exponential geodesic kernel are appreciated.

**Weaknesses**:

**Weak experiments; unclear if advances are practically relevant**

The experiments in this paper consist of (1) qualitative uncertainty visualizations for two selected features, (2) showing that uncertainty decreases for regularized models. Either of these experiments are not convincing demonstrations of the usefulness of uncertainty quantification, as they do not deliver on the promise of "whether or not to trust an explanation". In particular, it is still unclear whether there is a clear, concrete setting where the proposed method is truly advantageous. Some examples of experiments closer to this goal include, (1) studying the faithfulness or robustness of explanations, and comparing them with this uncertainty metric. Does the proposed metric provide a cheaper way to identify data points whose explanations are likely to be unreliable based on these (or other) metrics? (2) A toy example where the explanations and the underlying uncertainties associated with explanations are known in advance, and a demostration of whether the proposed approach is able to correctly recover these. Note that these are mere suggestions, and that any experiment that can convince the readers of potential usefulness and practical utility of these methods is welcome.

**Missing discussion of scalability and statistical efficiency of proposed approach**

The core components of the proposed algorithm involve sampling from the decision boundary and computation of the geodesic distance. These are computationally expensive for large problems, as due to the curse of dimensionality one would expect the cost of characterizing the decision boundary (by sampling) to be exponential in the data dimensionality. The authors claim that they "apply adversarial attacks to reduce search space size" for MNIST, however it is unclear if this heuristic results in generation of independent samples from the decision boundary as required for the monte carlo sampling approach. Some discussion on the statistical properties of this approach and its computational complexity would improve the paper in my view.

**Distance from DB may not be as important as its curvature**

Consider the case of "explaining" a linear model, where some methods (gradients, smoothgrad) return the weight vector itself as an explanation (https://arxiv.org/abs/1711.06104), and thus there is no uncertainty in this case. However the proposed method still assigns non-zero uncertainty values to explanations based on both the geodesic distance as well as distance from the DB, which seems to contradict the ground truth. Thus the proposed approach may not be ideal to quantify uncertainty, and some discussion on this counterexample is welcome.


**Summary Of The Paper:**

The paper proposes to quantify uncertainty of explanations by modelling explanations via a Gaussian process regression, and by defining a kernel called the "weighted exponential geodesic" (WEG) kernel. This kernel computes the similarity between two data points via their distances to the decision boundary, and the geodesic distance (of their "projections") on the decision boundary. Experiments show that the proposed measure indeed places larger uncertainty close to the decision boundary, and smaller uncertainty elsewhere.

**Summary Of The Review:**

While the proposed WEG kernel is interesting, the experiments are underwhelming, and do not convincingly demonstrate that the proposed approach achieves its goals of telling users when to trust an explanation method. As a result, I am leaning toward a reject.

---

> ### Author Response · Authors · 2022-11-17
> **Response 1/2**
>
> Thank you for your constructive review – your detailed feedback is very helpful for improving our paper. We have provided responses below to the points that you have raised, which we hope will help provide additional clarification. We have also made a number of changes to the manuscript based on your feedback (see the “general response” comment posted separately). Please feel free to ask any other followup questions.
>
> ---
>
> **Q1. Weak Experiments and Practical Relevance**
>
> **A1.** We have added a toy example in Appendix F.2 ([Github Link](https://github.com/anonymousGPEC/GPEC/blob/main/Figures/Figure6.png)) comparing GPEC uncertainty estimates on a linear model and a more complicated function. In the bottom row of the figure we show a gradient-based estimate of feature importance for both models. We see that even using this deterministic feature importance method, the estimates can fail to be robust due to the nonlinearity (i.e., nearby samples within the same class can have very different explanations). Our experiments in section 5.2 also show that competing uncertainty methods also fail to capture this type of decision boundary-aware uncertainty, which we capture using the geodesic-based kernel in GPEC. With GPEC we can combine the decision boundary-aware uncertainty with the approximation uncertainty from the explainer which gives a more comprehensive estimate of uncertainty.
>
> Regarding practical relevance: feature attribution explanations are widely used among machine learning practitioners, however many of these methods have approximation variance or have issues with robustness (see related works in Section 2). Therefore it is important to understand when an explanation can be trusted. In the case of our toy example, the user should be made aware of the high uncertainty estimate for explanations near the black-box nonlinearity which could be potentially unstable.
>
> ---
>
> **Q2. Discussion on Scalability and Efficiency**
>
> **A2.** The main focus of our paper is to introduce the novel notion of decision boundary-aware uncertainty and develop a method, GPEC, to capture this uncertainty using the geometry of the decision boundary (DB). To the best of our knowledge, this is the first work using geodesic distances along the DB to characterize uncertainty from DB complexity. Algorithms for drawing samples from the decision boundary are still an active area of research. Due to the time complexity of DB estimation, this can result in a tradeoff between computation time and approximation accuracy. In our implementation for image datasets we use the DeepDIG algorithm from [1], which proposes the adversarial example heuristic. While this is only an approximation of the DB, we note that GPEC only needs to sample the local DB around the target explanation. This is due to the weighting function $\exp[-\rho || x-M||_2^2]$ (Eq. 5), which approaches zero as the distance to the DB increases. Therefore given that the test samples are from the same distribution as the training distribution, the DeepDIG algorithm should give a reasonable approximation of the local DB. We have added this clarification to the discussion section.
>
> Regarding the time complexity of GPEC, because GPEC has separate training and inference steps, generating uncertainty estimates for new explanations can be significantly faster than competing methods such as BayesLIME and BayesSHAP. We show an inference time comparison in App. F Table 2 for reference ([Github Link](https://github.com/anonymousGPEC/GPEC/blob/main/Figures/Table2.png)). Time complexity for predictions in exact Gaussian Process models is typically $\mathcal{O}(n^3)$, where $n$ is the number of explanation samples selected to train GPEC. This complexity can also be further reduced using (e.g.) variational methods.

---

> > ### Author Response · Authors · 2022-11-17
> > **Response 2/2**
> >
> >
> > **Q3. Distance from Decision Boundary compared to Decision Boundary Curvature**
> >
> > **A3.** We agree that the distance from the DB is not as important as geometric properties of the decision boundary itself. We characterize the complexity of the DB using geodesic distances along the decision boundary. For example, given two fixed points $x,y \in \mathbb{R}^2$, the line segment with minimum geodesic distance connecting the $x$ and $y$ will have zero curvature (i.e. a straight line). Therefore a linear model would have the least complex decision boundary under this characterization, and thus sample explanations should have the lowest possible uncertainty under our framework. We capture geodesic distances between decision boundary samples using the Exponential Geodesic (EG) kernel (Eqn. 4).
> >
> > However, since explanations are typically not directly on the decision boundary, we relate the explanations to the EG kernel through a weighting based on their distance to the decision boundary. Specifically, we weight each element of the EG kernel matrix based on the distance of the decision boundary samples to the explanation, which results in the WEG kernel similarity (Eqn. 5). Therefore the distances of each explanation to the decision boundary is only used for weighting purposes; the WEG kernel primarily captures the complexity of the decision boundary through geodesic distances.
> >
> > Regarding the case of explaining a linear model with a gradient-based explainer, since the explanation has no inherent variance we would ignore the function approximation uncertainty component of GPEC (see Eqn. 2). GPEC would still capture decision boundary-aware uncertainty, which would be relatively constant for a linear model (see Figure 6 in Appendix F.2 ([Github Link](https://github.com/anonymousGPEC/GPEC/blob/main/Figures/Figure6.png)) and represent the lowest uncertainty as compared to a nonlinear DB. Note that due to the training of  the GP model, there may still be differences in uncertainty estimates depending on the training distribution (e.g. for out-of-distribution test samples).
> >
> >  ---
> >
> > **Q4. Discussion of Theorem 2**
> >
> > **A4.** Thank you for this feedback; we have added additional explanation for Theorem 2 and its implications in the revision.
> >
> > ---
> >
> > **Q5. Reproducibility**
> >
> > **A5.** We have added the GPEC algorithm in Alg. 1 in Appendix D ([Github Link](https://github.com/anonymousGPEC/GPEC/blob/main/Figures/Algorithm1.png)) and added more detail in the implementation section. We hope this helps to clarify the overall process. We have also added a link to the anonymized code in a separate response, which will be made public when the paper is accepted.

---

> > > ### Comment · Reviewer_dnzT · 2022-11-26
> > > **Rebuttal response**
> > >
> > > Thank you for your rebuttal, and for the discussion on the case of explaining a linear model.
> > >
> > > Regarding Q1., the uncertainty in your toy example arises entirely due to model non-linearity. In that case, why not simply measure the model's Hessian norm at that point to quantify the extent of non-linearity, and use this as a metric of whether or not to trust the model?
> > >
> > > Regarding Q3., it seems that the weighting is indeed primarily responsible for the non-zero uncertainty estimates obtained for linear models. This hints at the fact that the proposed method may not be suitable to capture uncertainty, as it fails to capture the appropriate notion of uncertainty for the toy example of linear models.
> > >
> > > Overall, the original objections of weak experiments, and unclear practical relevance remain, and I shall retain my current score.

---

> > > > ### Author Response · Authors · 2022-12-03
> > > > **Clarification**
> > > >
> > > >
> > > > Thank you for your response. We want to briefly clarify the points that you mentioned.
> > > >
> > > > ---
> > > >
> > > > **Regarding Q1.** We previously considered using the Hessian, however a Hessian-based approach has a number of limitations:
> > > >
> > > > a) A Hessian-based approach would severely restrict the type of black-box models you can use with GPEC. One of the main applications of GPEC is on tabular data where non-differentiable models (e.g. Random Forest, XGBoost) are commonly used; requiring the Hessian would preclude this class of models from being used with GPEC.
> > > >
> > > > b) The Hessian does not necessarily correspond to model complexity. For example, the Hessian norm of a piecewise linear model (e.g. ReLU network) would be zero everywhere but the model can be made to be arbitrarily complex.
> > > >
> > > > c) The Hessian is an infinitesimally local pointwise estimate and can therefore fail to capture nearby nonlinearities. For example, consider the function $f(x) = \sin(\beta x)$ with some $ \beta  \in \mathbb{R}$.  The second derivative evaluated at $x = \frac{2 \pi}{\beta}$ is zero, which would suggest linearity. However, the second derivative at the point $\tilde x = x + \epsilon$ for any $\epsilon>0$ is $-\beta^2 \sin(2 \pi + \beta \epsilon)$, which is unbounded w.r.t. $\beta$. In this example, the second derivative could indicate no uncertainty at $x$ (since $f’’(x) = 0$) and extremely high uncertainty at $\tilde x$, despite the two points $x$ and $\tilde x$ being arbitrarily close together.
> > > >
> > > > In contrast, our proposed geodesic-based kernel similarity is a) model-agnostic, b) corresponds to decision boundary complexity (Theorem 2), and c) captures the complexity of the decision boundary over a neighborhood around the points which is controlled by the weighting.
> > > >
> > > > ---
> > > >
> > > > **Regarding Q3.**  I think there is a misunderstanding with the linear toy example. When using the GPEC framework we would expect to see a relatively small and consistent uncertainty value for explanations as compared with a more complex model. This is because GPEC combines two sources of uncertainty: (1) DB-aware uncertainty, and (2) Function Approximation uncertainty. From Eqn. 1:
> > > >
> > > > $e_m = \mathcal{E}(x_m) + \eta_m$
> > > >
> > > > $\underbrace{\mathcal{E}(x_m) \sim \mathcal{GP}(0,k(x, x'))}_{\textrm{1. Decision Boundary-Aware Uncertainty}}$
> > > >
> > > > $\underbrace{\eta_m \sim \mathcal{N}(0,\tau_m^{-1}  )}_{\textrm{2. Function Approximation Uncertainty}}$
> > > >
> > > > where $e_m$ is a sampled explanation. Since the gradient explanation has no variance, the function approximation uncertainty (2) would be zero. This is consistent with the intuition that there is no approximation uncertainty for the gradients of the linear model.
> > > >
> > > > However, the benefit and novelty of GPEC is that we additionally derive DB-aware uncertainty (1), which is consistent with a Bayesian perspective on uncertainty. For a given explanation, we want to understand the behavior of nearby explanations w.r.t. the decision boundary of the prediction model. Therefore, without knowing the entirety of the explanation space (i.e., without infinite samples), there can still be uncertainty. Note that given infinite samples over the explanation space, the DB-aware uncertainty becomes zero for the linear example.
> > > >
> > > > Therefore in the GPEC uncertainty estimate it is more important to ensure that the linear model achieves minimum uncertainty as compared with nonlinear models. The linear toy example validates that this property holds, and the results are consistent with the expected uncertainty estimate from GPEC.

---

### Official Review · Reviewer_6BBB · 2022-10-24

**Confidence:** 3
**Correctness:** 3
**Technical Novelty And Significance:** 3
**Empirical Novelty And Significance:** 2
**Recommendation:** 5

**Clarity, Quality, Novelty And Reproducibility:**

This paper is well-presented and organized. The proposed idea is novel. The authors provide the code for experiment result reproduction.


**Strength And Weaknesses:**

Strengths:
1. The authors introduce a geometric perspective on capturing explanation uncertainty and define a novel geodesic-based similarity between explanations.
2. The authors prove theoretically that the proposed similarity captures the complexity of the decision boundary from a given black-box classifier.
3. This paper proposed a novel Gaussian Process-based framework.


Weaknesses:
1.  Since the proposed method is a combination of several components, an ablation study is necessary to identify the contribution of each component. Otherwise, it is difficult to evaluate the effectiveness of the proposed method.

2. Performance is lacking on large-scale benchmarks. It is better to show some experimental results in large-scale datasets (e.g., CIFAR100, ImageNet) to demonstrate the effectiveness of the proposed method.

3. Some important related work missing, such as [1, 2]

[1] Zhang, Yujia, Kuangyan Song, Yiming Sun, Sarah Tan, and Madeleine Udell. "Why Should You Trust My Explanation?" Understanding Uncertainty in LIME Explanations." arXiv preprint arXiv:1904.12991 (2019).
[2] Patro, Badri N., Mayank Lunayach, Shivansh Patel, and Vinay P. Namboodiri. "U-cam: Visual explanation using uncertainty based class activation maps." In Proceedings of the IEEE/CVF International Conference on Computer Vision, pp. 7444-7453. 2019.


**Summary Of The Paper:**

This paper focuses on the problem of reliable explanations. The authors find that existing methods can be inconsistent or unstable. Therefore, there is an impending need to quantify the uncertainty of such explanation methods in order to understand when explanations are trustworthy. The authors introduce a novel uncertainty quantification method parameterized by a Gaussian Process model, which combines the uncertainty approximation of existing methods with a novel geodesic-based similarity which captures the complexity of the target black-box decision boundary.


**Summary Of The Review:**

See *Strength And Weaknesses*

---

> ### Author Response · Authors · 2022-11-17
> **Response**
>
> Thank you for your feedback and for appreciating the novelty of our results. We have provided responses below to the points that you have raised, which we hope will help provide additional clarification. We have also made a number of changes to the manuscript based on your feedback (see the “general response” comment posted separately). Please feel free to ask any other followup questions.
>
> ---
>
> **Q1. Ablation Study**
>
> **A1.** GPEC combines two different sources of uncertainty: 1) the decision boundary-aware uncertainty and 2) function approximation uncertainty from the explainer. For 1), we compare GPEC using the proposed decision boundary-aware WEG kernel (GPEC-WEG) and GPEC using the RBF kernel (GPEC-RBF). This comparison is shown in columns 1 and 2 in Figure 2 where we see that the uncertainty from GPEC-RBF does not capture decision boundary information.
>
> For 2), we compare GPEC-WEG with and and without function approximation uncertainty in Figure 5 (Row B and Row A, respectively). Row C in Figure 5 shows the change in uncertainty from including the function approximation uncertainty.
>
> We thank the reviewer for their feedback and have included clarification in the experiments section pointing out these comparisons.
>
> ---
>
> **Q2. Large-Scale Benchmarks**
>
> **A2.** The main focus of our paper is to introduce the novel notion of decision boundary-aware uncertainty and develop a method, GPEC, to capture this uncertainty using the geometry of the decision boundary (DB). To the best of our knowledge, this is the first work using geodesic distances along the DB to characterize uncertainty from DB complexity. We reserve the optimization and performance aspect of GPEC to future extensions. There is not enough time during the response period to add larger datasets, however we will add results for CIFAR10/100 in the camera-ready version.
>
> ---
>
> **Q3. Related Works**
>
> **A3.** Thank you for pointing out these additional related works. We have added them to the revision.

---

### Official Review · Reviewer_R7Fs · 2022-10-27

**Confidence:** 3
**Correctness:** 4
**Technical Novelty And Significance:** 3
**Empirical Novelty And Significance:** 2
**Recommendation:** 5

**Clarity, Quality, Novelty And Reproducibility:**

The paper is good/nice to read and ok to follow, but somehow it's not exactly clear, which impacts in particular the reproducibility. I think the idea of using the geodesic distance on the decision boundary is sound and its application in this context seems novel.

**Strength And Weaknesses:**

# Strengths
* The model choices are generally well substantiated
* The authors explain well the (computational) challenges and limits of the method and possible remedies
* The figures are helpful to illustrate the ideas
* The proposed idea to use the geodesic kernel in the Gaussian process is sound and quite neat
# Weaknesses
* Although I feel like I understand the idea well and the reasoning behind it, I have problems to really understand the whole training and inference process. There is this one paragraph summarizing the method, but it depicts the method only textually and I am missing a more technical description of the method. After all, a lot of things come here together.
* I am sorry, but I have to ask: do we really need extra methods to derive uncertainty for explaining models of the actual model? It appears to me that the explainer is not really needed that much here after all, only for the variance term. Can't the proposed method be used as a confidence measure of the black-box model itself? There is a whole subsection describing well the issues with explaining models, their insufficiency, and their lack of actually explaining what is happening in the model. Why would we want to fix these flawed models, instead of scraping them and doing something proper? The proposed method itself relies on multiple hyper-parameters and it introduces another layer of possible inaccuracies, introducing possibly another false sense of trust. I believe that when the model could be used at least directly to replace the absolute failing uncertainty measure of softmax, that this would be a good contribution. I wonder if this is possible or if the inference would then take too long. This brings me to the next point:
* The sample complexity seems to be high, especially when brittle decision boundaries are supposed to be detected. I approve that there is a complexity analysis, but I wonder if in practice the time to train and to conduct inference for this model is so high that it prevents the usage of big datasets.

**Summary Of The Paper:**

The authors propose a model that estimates the confidence of an explanation model of a classification model. The proposed model takes into account the variance in explanations of the explanation model, as well as the brittleness of the decision boundary around the sample that is to be explained. Samples that lie around a _complex_ decision boundary are assigned a high uncertainty. The complexity of the decision boundary is here measured as the length of the shortest path on the decision boundary that connects two close samples. This definition requires the computation of a geodesic distance, which is approximated over a Monte-Carlo sampling method. The experiments compare uncertainty methods for explanations on MNIST, fashion MNIST and three UCI datasets.

**Summary Of The Review:**

Nice idea with a somewhat flawed application, trying to build upon the flawed explaining models. Clarity should be improved in the rebuttal. Training and Inference complexity is possibly awfully high. Because I like the idea, I would argue for a weak reject, although I see many issues with the paper as is.

# After Rebuttal Thoughts
The authors accomodated my request for clarity regarding the final method. I still have my concerns about the proposed approach, because it creates another layer of trust, that is however not certified. This is my main issue with the paper. The issue with the inference complexity also seems to consist, although I truly appreciate that the authors added a discussion about this.

---

> ### Author Response · Authors · 2022-11-17
> **Response**
>
>
> Thank you for your feedback and for appreciating the novelty of our geodesic-based kernel. We have provided responses below to the points that you have raised, which we hope will help provide additional clarification. We have also made a number of changes to the manuscript based on your feedback (see the “general response” comment posted separately). Please feel free to ask any other followup questions.
>
> ---
>
> **Q1. Technical description of training and inference process**
>
> **A1.** We have added the GPEC algorithm in Alg. 1 in Appendix D ([Github Link](https://github.com/anonymousGPEC/GPEC/blob/main/Figures/Algorithm1.png)) and added more detail in the implementation section. We hope this helps to clarify the process.
>
> ---
>
> **Q2. Motivation of training GPEC to estimate uncertainty of explanations**
>
> **A2.** The task we are specifically investigating is how to improve trustworthiness of the black-box model explanations. Feature attribution explanations are popular among machine learning practitioners, however many of these methods have approximation variance or have issues with robustness (see related works in Section 2). There are also many situations (as in our example in Section 1) where the model explanations are just as important as the model predictions. In these cases the user would want to know if there is uncertainty in the explanation estimate in order to highlight the limitations of the estimate.This uncertainty can arise from 1) approximation of the explainer function or 2) nearby nonlinearities in the decision boundary of the black-box model. To the best of our knowledge, this is the first work to characterize explanation uncertainty in this way and to introduce a method to combine these two sources of uncertainty.
>
> We agree that existing explanation methods are flawed with respect to the decision boundary-aware uncertainty we introduce, but often the choice of feature attribution method is dependent on or limited by the application at hand. GPEC allows for the user to select the best feature attribution explainer for their application and still obtain an uncertainty estimate that includes the decision boundary-aware uncertainty. We also believe that publishing this work and its novel characterization of explanation uncertainty would help motivate future extensions in fixing the flawed explainer models.
>
> The explainer is required when using GPEC because we use the explanations as labels for the underlying Gaussian Process (GP) model. We also take the functional approximation uncertainty estimate directly from the explainer. By using the geodesic-based WEG kernel, we can then take into account the distribution of explanations and increase the uncertainty of explanations near nonlinearities in the decision boundary. We appreciate your suggestion of using GPEC to directly estimate uncertainty of the black-box model predictions; this is an interesting future direction that could also benefit from the geodesic-based approach of estimating decision boundary complexity.
>
>
>
> ---
>
> **Q3. Sample complexity seems high**
>
> **A3.** The main focus of our paper is to introduce the novel notion of decision boundary-aware uncertainty and develop a method, GPEC, to capture this uncertainty using the geometry of the decision boundary. We reserve the optimization and performance aspect of GPEC to future extensions.
>
> However, because GPEC has separate training and inference steps, generating uncertainty estimates for new explanations can be significantly faster than competing methods such as BayesLIME and BayesSHAP. We show an inference time comparison in App. F Table 2 for reference ([Github Link](https://github.com/anonymousGPEC/GPEC/blob/main/Figures/Table2.png)), and we compare favorably to competing methods. Time complexity for predictions in exact Gaussian Process models is typically $\mathcal{O}(n^3)$, where $n$ is the number of explanation samples selected to train GPEC. This complexity can also be further reduced using (e.g.) variational methods. GPEC training cost is indeed constrained by the decision boundary estimation step, which is an active area of research. We have added this additional discussion on sample complexity to the revised paper.

---

### Author Response · Authors · 2022-11-17
**General Comment**

Thanks to all the reviewers for reading our work and providing feedback. We will respond separately for each reviewer’s specific comments. After considering the provided feedback, we have made the following changes to the paper which are highlighted in blue in the uploaded revision:

* Added anonymized code (Posted in a separate response below)
* Added GPEC algorithm (Alg. 1 in Appendix D)
* Expanded the explanation of Theorem 2 to improve clarity and intuition.
* Added discussion for a case study of applying GPEC on a linear model (Appendix F.2).
* Improved clarity of Figure 4 and Figure 5 with respect to ablation study of GPEC components.
* Added related works [1] and [2]

---

[1] Zhang, Yujia, Kuangyan Song, Yiming Sun, Sarah Tan, and Madeleine Udell. "Why Should You Trust My Explanation?" Understanding Uncertainty in LIME Explanations." arXiv preprint arXiv:1904.12991 (2019).

[2] Patro, Badri N., Mayank Lunayach, Shivansh Patel, and Vinay P. Namboodiri. "U-cam: Visual explanation using uncertainty based class activation maps." In Proceedings of the IEEE/CVF International Conference on Computer Vision, pp. 7444-7453. 2019.

---

### Decision · Program_Chairs · 2023-01-20

**Decision:**

Reject

**Justification For Why Not Higher Score:**

N/A

**Justification For Why Not Lower Score:**

N/A

**Metareview: Summary, Strengths And Weaknesses:**

This paper proposes to quantify uncertainty of explanations by modelling explanations via a Gaussian process regression, and by defining a kernel called the "weighted exponential geodesic" (WEG) kernel. The reviewers and AC agree that the lack of stronger evidence for the effectiveness of this approach calls into question its practical relevance and usefulness. Further research is necessary to strengthen the case for this method and demonstrate its ability to achieve its intended objective.